# Multiple 9-1-1 complexes promote homolog synapsis, DSB repair, and ATR signaling during mammalian meiosis

**Catalina Pereira[1], Gerardo A Arroyo-Martinez[1], Matthew Z Guo[1], Michael S Downey[1], Emma R Kelly[2], Kathryn J Grive[3], Shantha K Mahadevaiah[4], Jennie R Sims[5], Vitor M Faca[6], Charlton Tsai[1], Carl J Schiltz[1], Niek Wit[7], Heinz Jacobs[7], Nathan L Clark[8], Raimundo Freire[9,10,11], James Turner[4], Amy M Lyndaker[2], Miguel A Brieno-Enriquez[12], Paula E Cohen[1], Marcus B Smolka[5], Robert S Weiss[1]***

[1]Department of Biomedical Sciences, Cornell University, Ithaca, United States; [2]Division of Mathematics and Natural Sciences, Elmira College, Elmira, United States; [3]Department of Obstetrics and Gynecology, Brown University, Providence, United States; [4]Sex Chromosome Biology Laboratory, The Francis Crick Institute, London, United Kingdom; [5]Department of Molecular Biology and Genetics, Weill Institute for Cell and Molecular Biology, Cornell University, Ithaca, United States; [6]Department of Biochemistry and Immunology, FMRP, University of São Paulo, Ribeirão Preto, Brazil; [7]Division of Immunology, The Netherlands Cancer Institute, Amsterdam, Netherlands; [8]Department of Human Genetics, University of Utah, Salt Lake City, United States; [9]Unidad de Investigación, Hospital Universitario de Canarias, Tenerife, Spain; [10]Instituto de Tecnologías Biomédicas, Universidad de La Laguna, La Laguna, Spain; [11]Universidad Fernando Pessoa Canarias, Las Palmas de Gran Canaria, Spain; [12]Magee-Womens Research Institute, Department of Obstetrics, Gynecology and Reproductive Sciences, University of Pittsburgh, Pittsburgh, United States

*For correspondence:
rsw26@cornell.edu

**Abstract** DNA damage response mechanisms have meiotic roles that ensure successful gamete formation. While completion of meiotic double-strand break (DSB) repair requires the canonical RAD9A-RAD1-HUS1 (9A-1-1) complex, mammalian meiocytes also express RAD9A and HUS1 paralogs, RAD9B and HUS1B, predicted to form alternative 9-1-1 complexes. The RAD1 subunit is shared by all predicted 9-1-1 complexes and localizes to meiotic chromosomes even in the absence of HUS1 and RAD9A. Here, we report that testis-specific disruption of RAD1 in mice resulted in impaired DSB repair, germ cell depletion, and infertility. Unlike *Hus1* or *Rad9a* disruption, *Rad1* loss in meiocytes also caused severe defects in homolog synapsis, impaired phosphorylation of ATR targets such as H2AX, CHK1, and HORMAD2, and compromised meiotic sex chromosome inactivation. Together, these results establish critical roles for both canonical and alternative 9-1-1 complexes in meiotic ATR activation and successful prophase I completion.

## Editor's evaluation

This paper nicely provides the roles of 9-1-1 checkpoint clamps (three DNA damage response clamps; Rad9A-Rad1-Hus1, Rad9B-Rad1-Hus1, Rad9B-Rad1-Hus1B) in mouse male meiosis, particularly DSB repair, chromosome synapsis, checkpoint signaling, and meiotic sex chromosome silencing.

## Introduction

DNA damage response (DDR) mechanisms protect genomic integrity by sensing and repairing DNA lesions or initiating apoptosis when lesions are unrepairable (*Blackford and Jackson, 2017*). DDR proteins are also essential for successful haploid gamete formation. Although double-strand DNA breaks (DSBs) are considered to be the most toxic form of DNA damage, meiotic recombination relies on SPO11-induced DSBs for homologous chromosomes to synapse, exchange genetic material, and properly segregate at the first meiotic division (*Bolcun-Filas et al., 2014*; *Gray and Cohen, 2016*). Of particular importance are the meiotic events that occur during the five substages of prophase I, a major feature of which involves the transient formation of the proteinaceous structure called the synaptonemal complex (SC) (*Cahoon and Hawley, 2016*; *Gray and Cohen, 2016*). During the first stage, leptonema, axial elements containing SC protein 3 (SYCP3) form along condensed chromosomes (*Page and Hawley, 2004*). Additionally, the DNA damage marker, γH2AX, accumulates during leptonema as chromosomes experience SPO11-induced DSBs. Progression into zygonema is characterized by the pairing and synapsis of chromosomes, marked by the presence of the central element protein SC protein 1 (SYCP1). During pachynema, DSB repair is completed, and by mid-pachynema γH2AX is no longer present on the fully synapsed autosomes. However, in male meiocytes, abundant γH2AX is apparent at the sex body containing the X and Y chromosomes, which synapse only in a small domain called the pseudoautosomal region but otherwise remain unsynapsed. Meiotic cells subsequently enter diplonema, featuring dissolution of the central element while homologous chromosomes remain tethered by crossovers. Breakdown of the SC marks the final stage in prophase I, diakinesis.

Ataxia-telangiectasia and Rad3-related (ATR) kinase is a key regulator of recombinational DSB repair and synapsis throughout meiotic prophase I (*Pereira et al., 2020*). ATR activation in somatic cells has been well characterized; however, the mechanisms of meiotic ATR activation have not been fully elucidated. ATR activation in response to replication stress and other signals in mitotic cells is known to involve interaction between the RAD9A-RAD1-HUS1 (9A-1-1) complex and topoisomerase 2-binding protein I (TOPBP1) (*Blackford and Jackson, 2017*). The toroidal, PCNA-like 9A-1-1 complex is loaded at recessed DNA ends by the RAD17–replication factor C (RFC) clamp loader (*Eichinger and Jentsch, 2011*). ATR in association with ATR interacting protein (ATRIP) independently localizes to replication protein A (RPA)-coated single-stranded DNA (*Zou and Elledge, 2003*). The 9A-1-1 complex then interacts with RAD9A-RAD1-HUS1 interacting nuclear orphan (RHINO) and TOPBP1, which allows TOPBP1 to activate ATR via its ATR-activating domain (*Cotta-Ramusino et al., 2011*; *Delacroix et al., 2007*; *Lindsey-Boltz et al., 2015*). ATR activation initiates several downstream processes such as cell cycle arrest, DNA repair, fork stabilization, and inhibition of new origin firing, or triggers apoptosis (*Saldivar et al., 2017*). Independent of 9A-1-1/TOPBP1, ATR also can be directly activated during a normal mitotic cell cycle by Ewing's tumor-associated antigen 1 (ETAA1), in part to promote metaphase chromosome alignment and spindle assembly checkpoint function (*Bass and Cortez, 2019*).

During meiotic prophase I, homologous chromosomes pair and undergo recombination, with regions of asynapsis being subjected to DDR-dependent transcriptional silencing. ATR, along with meiosis-specific HORMA (Hop1, Rev7, and Mad2)-domain proteins, TOPBP1, and other factors, localizes to unsynapsed chromatin regions in leptotene- and zygotene-stage cells (*Fedoriw et al., 2015*; *Keegan et al., 1996*; *Kogo et al., 2012*; *Perera et al., 2004*; *Shin et al., 2010*; *Wojtasz et al., 2009*). At pachynema, the homologs are fully synapsed, at which point ATR localizes only to the unsynapsed axes and throughout the chromatin of the X and Y chromosomes, where it triggers a mechanism called meiotic sex chromosome inactivation (MSCI). MSCI is essential for successful meiotic progression through the silencing of toxic Y-linked genes and sequestration of DDR proteins away from autosomes (*Abe et al., 2020*; *Royo et al., 2010*; *Turner, 2015*; *Turner et al., 2006*). Central to MSCI is ATR-dependent recruitment of BRCA1 and other factors, and subsequent spreading of H2AX phosphorylation via the adaptor MDC1 (*Ichijima et al., 2011*; *Royo et al., 2013*; *Turner et al., 2004*). Similarly, ATR mediates meiotic silencing of unsynapsed chromatin (MSUC) at autosomes that have failed to synapse properly (*Turner, 2007*; *Turner, 2015*). Beyond silencing, ATR has an essential role in promoting RAD51 and DMC1 loading to enable meiotic DSB repair (*Pacheco et al., 2018*; *Widger et al., 2018*). Previous work indicates that HUS1 and RAD9A are largely dispensable for meiotic ATR activation (*Lyndaker et al., 2013a*; *Vasileva et al., 2013*), raising the intriguing possibility

that HUS1B- and RAD9B-containing alternative 9-1-1 complexes contribute to ATR activation during mammalian meiosis.

In addition to its ATR-activating role, the 9A-1-1 complex also functions as a molecular scaffold for proteins in multiple DNA repair pathways. For example, the 9A-1-1 complex participates in homologous recombination by interacting with the RAD51 recombinase (*Pandita et al., 2006*) and EXO1 exonuclease (*Karras et al., 2013*; *Ngo et al., 2014*; *Ngo and Lydall, 2015*). Consistent with these observations from mitotic cells, RAD9A co-localizes with RAD51 on meiotic chromosome cores (*Lyndaker et al., 2013a*), and RAD1 similarly co-localizes with DMC1 when visualized by immuno-fluorescence staining, although higher-resolution immunoelectron microscopy analysis indicates that RAD1 and DMC1 are at distinct sites along meiotic chromosomes (*Freire et al., 1998*). In wild-type pachytene-stage cells, RAD51 foci are lost as DSBs are resolved, whereas without *Hus1* RAD51 is retained on spermatocyte autosomes into late prophase I (*Lyndaker et al., 2013a*).

Loss of any canonical 9A-1-1 subunit in mice leads to embryonic lethality (*Han et al., 2010*; *Hopkins et al., 2004*; *Weiss et al., 2000*). In conditional knockout (CKO) models, loss of *Hus1* or *Rad9a* in the testis results in persistent DSBs during meiotic prophase, leading to reduced testis size, decreased sperm count, and subfertility (*Lyndaker et al., 2013a*; *Vasileva et al., 2013*). Interestingly, the distribution of RAD1 and RAD9A on meiotic chromosome cores only partially overlaps, with RAD1 localizing as puncta on autosomes and coating asynapsed autosomes and the XY cores, and RAD9A present in a punctate pattern suggestive of DSB sites on autosomes and sex chromosomes (*Freire et al., 1998*; *Lyndaker et al., 2013a*). Although RAD9A fails to localize to meiotic chromosome cores in *Hus1*-deficient meiocytes, RAD1 localization is largely HUS1-independent, supporting the idea that RAD1 can act outside of the canonical 9A-1-1 complex.

The HUS1 and RAD9A paralogs, HUS1B and RAD9B, are highly expressed in testis (*Dufault et al., 2003*; *Hang et al., 2002*). Based on our previous results and the findings discussed above, we hypothesized that meiocytes contain alternative 9-1-1 complexes, RAD9B-RAD1-HUS1 (9B-1-1) and RAD9B-RAD1-HUS1B (9B-1-1B) (*Lyndaker et al., 2013b*). Since RAD1 has no known paralogs, it is expected to be common to both canonical and alternative 9-1-1 complexes. In order to elucidate the roles of each of the 9-1-1 complexes in mammalian meiosis, we generated *Rad1* CKO mice in which *Rad1* was disrupted specifically in male spermatocytes. *Rad1* CKO mice exhibited reduced sperm count, reduced testis size, and severe germ cell loss associated with DSB repair defects, consistent with previous studies of HUS1 and RAD9A. However, homolog synapsis and MSCI, which were largely unaffected in *Hus1* or *Rad9a* CKO mice, also were disrupted by *Rad1* loss. Furthermore, impaired phosphorylation of ATR substrates in *Rad1* CKO meiocytes indicated that the canonical and alternative 9-1-1 complexes work in concert to stimulate meiotic ATR signaling. This study highlights the importance of multiple 9-1-1 complexes during mammalian meiosis and establishes key roles for these DDR clamps in ATR activation, homolog synapsis, and MSCI.

## Results

### Evolution and tissue-specific expression of 9-1-1 subunits

Human RAD9A and RAD9B share 36% identity, while HUS1 and HUS1B are 48% identical (*Dufault et al., 2003*; *Hang et al., 2002*). Inspection of genomic sequences revealed that *Rad9b* genes are present in the syntenic genomic region of all placental species analyzed, whereas *Rad9a* was likely lost in a few species, including wallaby, tree shrew, and sloth (*Figure 1A and B*). Phylogenetic analysis suggested that the duplication event generating *Rad9a* and *Rad9b* occurred prior to the evolution of bony fish ancestors (*Danio rerio*), whereas the single-exon *Hus1b* gene likely arose after a retrocopy duplication event later in evolution in mammals. Ortholog matrix and evolutionary tree analyses of placental mammals further showed that *Rad1* is highly conserved, with no identifiable paralog.

Human and mouse gene expression data indicate that the 9-1-1 paralogs are highly expressed in testes but not other tissues, hinting at a potential role for RAD9B and HUS1B in spermatogenesis (*Figure 1—figure supplement 1A*). To further define the cell type-specific expression patterns of the 9-1-1 subunits within the testes, we mined single-cell RNA-sequencing data from wild-type adult mouse testis (*Grive et al., 2019*), comparing relative expression in spermatogonia, spermatocytes, and Sertoli cells (*Figure 1C*, *Figure 1—figure supplement 1B–D*). *Rad9b* expression was highest in spermatocytes as compared to spermatogonia and Sertoli cells. *Rad1* expression also was highest in

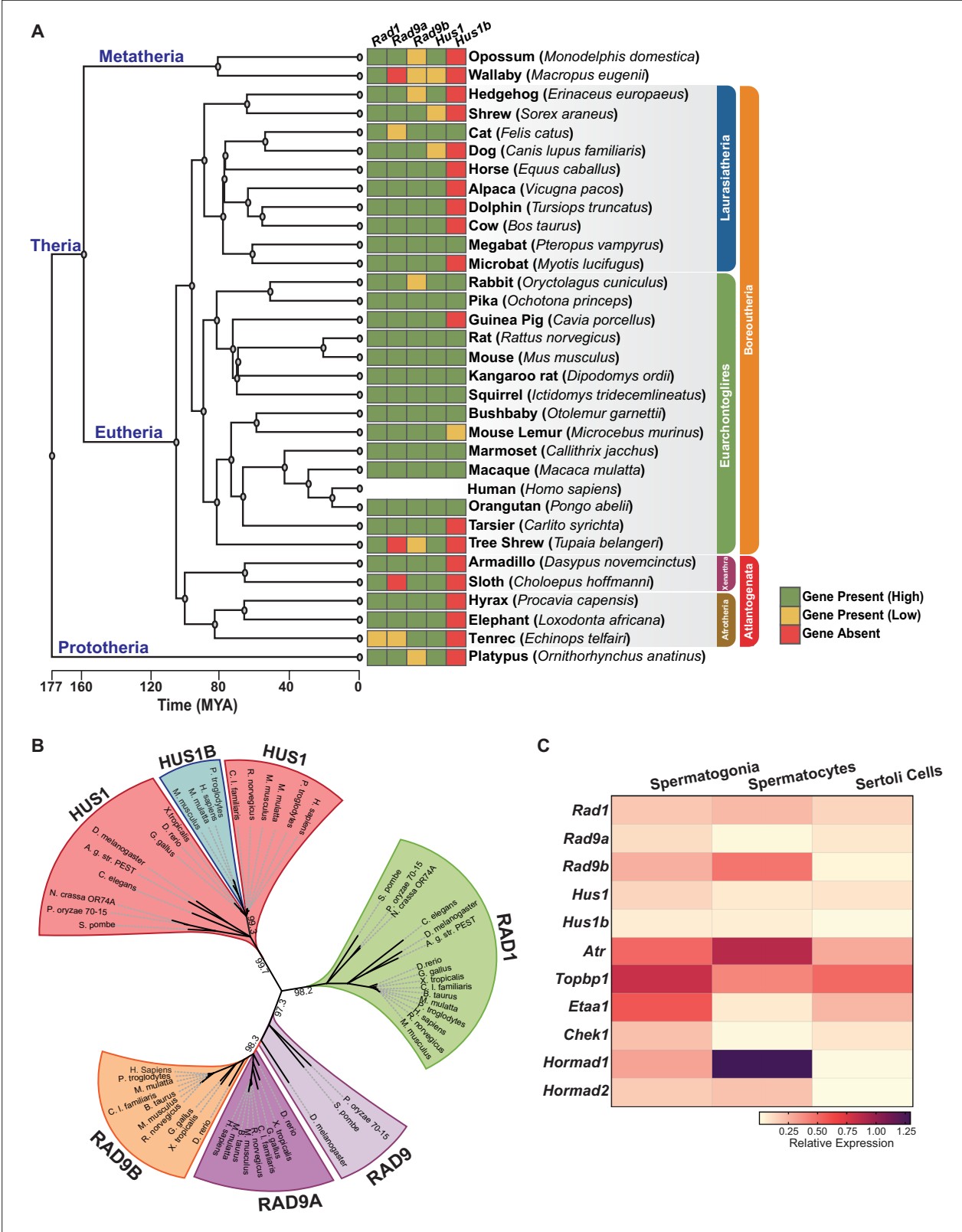

**Figure 1.** Phylogenetic analysis of 9-1-1 complex subunits. (**A**) Gene presence and absence matrix of human 9-1-1 subunit ortholog genes in 33 representative mammals. High confidence was determined if the genomic sequence had ≥50% of both target and query sequence identity, and a pairwise whole genomic alignment score of ≥50 when compared to human or if the genomic region containing the gene was syntenic with human. If an ortholog did not reach the threshold, then it was annotated as low confidence (yellow). If no ortholog was found, then it was considered absent

*Figure 1 continued on next page*

*Figure 1 continued*

(red). A cladogram was obtained from timetree.org. (**B**) Maximum likelihood unrooted phylogenetic tree of 9-1-1 subunit genes based on JTT + I + G + F. Protein sequences were obtained from NCBI HomoloGene and include bacteria (*Pleomorphomonas oryzae*), fungi (*Schizosaccharomyces pombe, Neurospora crassa*), nematode (*Caenorhabditis elegans*), true flies (*Drosophila melanogaster, Anopheles gambiae* str. *Pest*), fish (*Danio rerio*), frog (*Xenopus tropicalis*), bird (*Gallus gallus*), carnivora (*Canis lupus*), rodents (*Rattus norvegicus, Mus musculus*), and primates (*Homo sapiens, Mus musculus, Macaca mulatta, Pan troglodytes*). Sequences were aligned by Clustal Omega, and substitution model was tested on ProtTest. Ultrafast bootstrap (×1000 replicates) was performed in IQ-TREE web server, and nodes below 70% branch support were collapsed. Branch distance represents substitution rate. The lighter purple RAD9 denotes RAD9 prior to the duplication event. (**C**) Heatmap of single-cell RNA-sequencing data from mouse testes was queried to assess the expression of the indicated genes in spermatogonia, spermatocytes, and Sertoli cells. Expression of *Rad9b* in spermatocytes, p-value ≤ $5.47e^{-10}$, *Rad1* p-value ≤ $1.20e^{-09}$, *Hus1b* p-value ≤ $1.08e^{-08}$. Expression of *Rad9a* and *Hus1* in spermatogonia p-value ≤ $5.61e^{-19}$; p-value ≤ $3.78e^{-09}$. Relative expression is shown for each gene, with highest expression observed in purple and lowest expression observed in yellow.

The online version of this article includes the following source data and figure supplement(s) for figure 1:

**Source data 1.** Phylogenetic analysis of the 9-1-1 complexes.

**Source data 2.** Gene expression analysis in mouse testes.

**Source data 3.** tSNE plots showing single-cell RNA expression of 9-1-1 complex subunits in testes.

**Figure supplement 1.** Expression of 9-1-1 complex subunits.

spermatocytes, whereas *Hus1b* expression was similar in spermatocytes and spermatogonia. On the other hand, *Rad9a* and *Hus1* relative gene expression was highest in spermatogonia. As expected, expression of *Atr* and meiotic-silencing genes *Hormad1* and *Hormad2* was significantly higher in spermatocytes than spermatogonia or Sertoli cells, whereas *Etaa1* expression was relatively low in spermatocytes, consistent with prior reports that it has limited roles in meiosis (*Ellnati et al., 2017*). Spermatogonia also displayed relatively high levels of *Atr*, along with both *Topbp1* and *Etaa1*. Analysis of expression data from human testis showed that expression of the 9-1-1 paralogs *RAD9B* and *HUS1B* was highest in spermatocytes as compared to other testis cell types (Human Protein Atlas version 21.0 and *Guo et al., 2018*). Together, these results suggest that the 9-1-1/TOPBP1/ATR and ETAA1/ATR signaling axes are expressed in pre-meiotic spermatogonia and highlight potential roles for alternative 9-1-1 complexes in spermatocytes.

To further analyze the evolutionary relationships between 9-1-1 subunits, we performed evolutionary rate covariation (ERC) analysis, which assesses correlations in gene evolutionary history and can reveal functionally significant relationships (*Clark et al., 2012*; *Wolfe and Clark, 2015*). ERC analysis was performed between all of the 9-1-1 subunits in a pairwise fashion across 33 mammalian species. Significant ERC values were identified between the RAD1, HUS1, and RAD9B subunits, supporting the notion that alternative 9-1-1 complexes assemble in germ cells (*Figure 2A*). These findings are consistent with reports that RAD9B physically interacts with RAD1, HUS1, and HUS1B (*Dufault et al., 2003*), and similarly that HUS1B interacts with RAD1 (*Hang et al., 2002*), suggesting that the paralogs contribute to alternative 9-1-1 complexes that include RAD9B-RAD1-HUS1 (9B-1-1) and RAD9B-RAD1-HUS1B (9B-1-1B) (*Figure 2B*).

## Testis-specific RAD1 loss leads to increased germ cell apoptosis and infertility

To determine how disrupting the subunit shared by all of the 9-1-1 complexes impacted meiosis, we created a *Rad1* CKO model by combining a conditional *Rad1* allele (*Wit et al., 2011*) with *Stra8-Cre*, which drives CRE expression in spermatogonia (*Sadate-Ngatchou et al., 2008*). A similar approach was previously used to create *Hus1* CKO mice (*Lyndaker et al., 2013a*), also on the inbred 129Sv/Ev background, enabling direct comparison of results between the two models. Experimental *Rad1* CKO mice carried one *Rad1^flox^* allele, one *Rad1*-null allele, and *Stra8-Cre* (*Rad1^-/fl^; Cre^+^*). Mice that carried a wild-type *Rad1* allele (*Rad1^+/fl^; Cre^+^*) or lacked *Stra8-Cre* (*Rad1^-/fl^; Cre^-^* or *Rad1^+/fl^; Cre^-^*) were used as littermate controls. Both *Rad1* CKO and control mice were born at expected frequency.

Immunoblotting of whole testis lysates from adult (12-week-old) *Rad1* CKO mice confirmed significant reduction in RAD1 protein (*Figure 2C*). Reduced RAD1 expression was also observed in juvenile (postnatal day 14) *Rad1* CKO testes (*Figure 2—figure supplement 1*). The residual RAD1 protein observed in *Rad1* CKO mice may be attributed in part to somatic cells of the testis or pre-meiotic germ cells. However, additional results described below indicate that persistent RAD1 protein existed

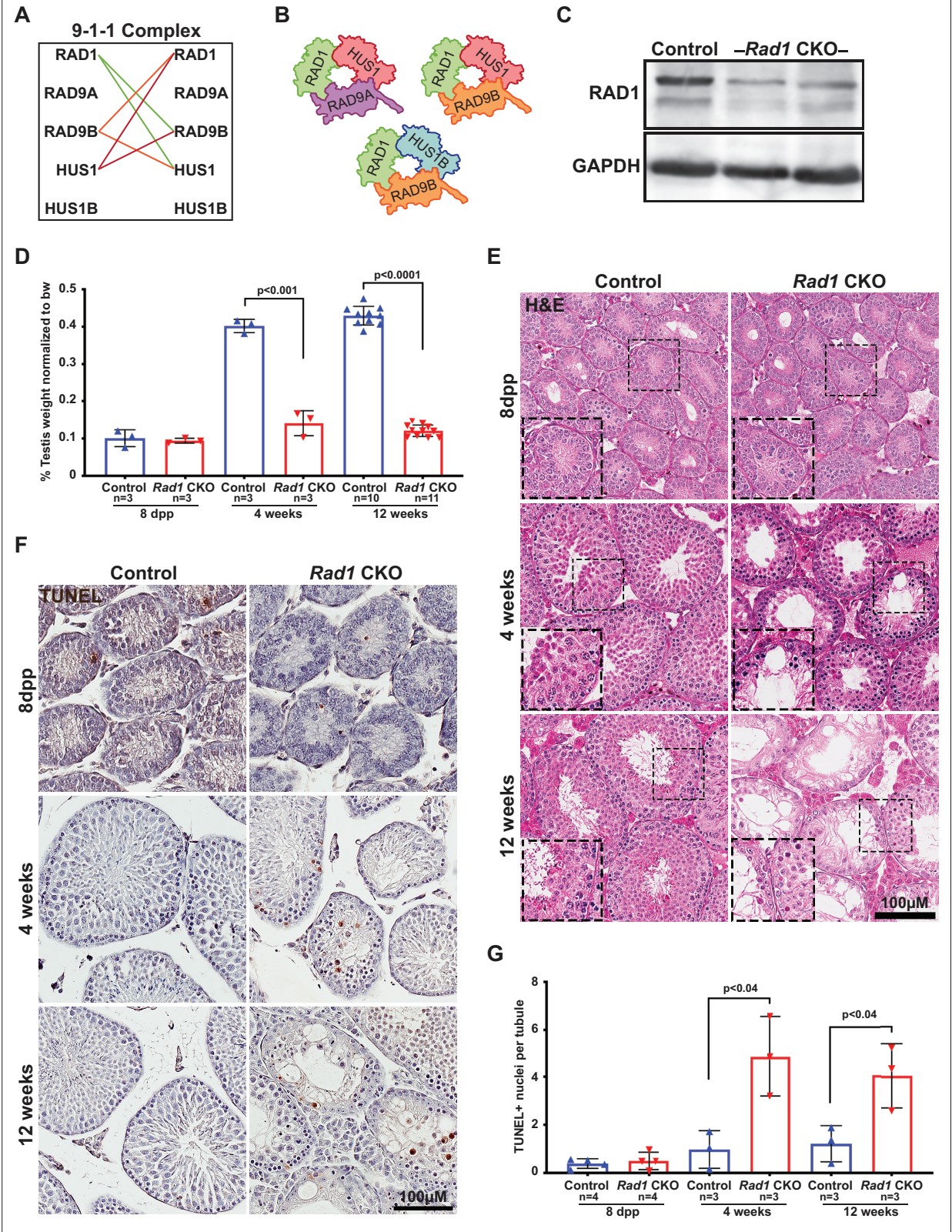

**Figure 2.** Conditional knockout (CKO) of the 9-1-1 complex subunit RAD1 causes severe germ cell loss in testes. (**A**) Evolutionary rate covariation analysis between 9-1-1 subunits. Lines depict significant covariance between 9-1-1 subunits. (**B**) Schematic showing putative meiotic 9-1-1 complexes: 9A-1-1, 9B-1-1, and 9B-1-1B. (**C**) Representative immunoblot for RAD1 in control and *Rad1* CKO whole testes lysates from 12-week-old mice (n = 5 control and 5 CKO samples analyzed in total). (**D**) Testis weight normalized to body weight from 8-day postpartum (dpp), 4-week-old, and 12-week-

*Figure 2 continued on next page*

*Figure 2 continued*

old control and *Rad1* CKO mice. (**E**) Seminiferous tubule cross sections from 8-dpp, 4-week-old, and 12-week-old mice were stained with H&E (representative images from three mice analyzed per age group per genotype). (**F, G**) Representative images (**F**) and quantification (**G**) of TUNEL-positive cells per tubule in control and *Rad1* CKO mice (50 tubules per mouse quantified; n = number of mice analyzed). p-Value calculated using Welch's unpaired *t*-test in GraphPad.

The online version of this article includes the following source data and figure supplement(s) for figure 2:

**Source data 1.** Control and *Rad1* conditional knockout (CKO) testes weights.

**Source data 2.** Control and *Rad1* conditional knockout (CKO) TUNEL+ cell counts.

**Source data 3.** Control and *Rad1* conditional knockout (CKO) TRA98+ cell counts.

**Source data 4.** Control and *Rad1* conditional knockout (CKO) LIN28+ cell counts.

**Figure supplement 1.** RAD1 levels are reduced in juvenile testes.

**Figure supplement 2.** *Rad1* inactivation in testis causes germ cell loss.

in some *Rad1* CKO spermatocytes due to partial CRE recombinase efficacy or perdurance of RAD1 protein from pre-meiotic stages. Testes from *Rad1* CKO males were one-third the size of control testes at 4 weeks of age, while body weight was not altered (***Figure 2D***). Hematoxylin and eosin (H&E) staining of testis sections from control and *Rad1* CKO mice showed a reduction in tubule size and cellularity starting at 4 weeks in CKO mice, with the phenotype being much more severe in 12-week-old mice (***Figure 2E***). Similar to previous findings in *Hus1* CKO males (***Lyndaker et al., 2013a***), histological analysis of *Rad1* CKO mice revealed increased apoptosis of zygotene/pachytene-stage cells. In *Rad1* CKO mice, round spermatids were infrequent but nevertheless observed in some histology sections from 4-week-old and 12-week-old mice, likely reflecting continued RAD1 expression in some meiocytes. Although severe germ cell loss in *Rad1* CKO mice prevented precise staging of seminiferous tubules, 65.7% ± 2.7% of tubules in *Rad1* CKO testes cross sections had fewer than 10 round spermatids, whereas no such tubules were identified in normal control testes (n = 3 mice per genotype; 50 tubules per mouse).

TUNEL staining confirmed significantly increased apoptosis in testes from *Rad1* CKO mice starting at 4 weeks of age (***Figure 2F and G***). 4-week-old *Rad1* CKO mice contained 2.4 ± 0.8 apoptotic nuclei per seminiferous tubule compared to 0.5 ± 0.4 in control mice. Apoptosis continued to be significantly elevated in 12-week-old *Rad1* CKO mice (2.0 ± 0.7 positive nuclei per tubule) as compared to control mice (0.6 ± 0.4 positive nuclei per tubule) and was apparent in zygotene/pachytene-stage cells (***Figure 2F***, ***Figure 2—figure supplement 2A***). To quantify the impact of *Rad1* loss on germ cells, we stained testis sections for the germ cell-specific antigen TRA98 (***Carmell et al., 2016***). Tubules from control mice at 4 or 12 weeks of age contained an average of 220.2 ± 26.3 or 254.3 ± 45.5 TRA98-positive cells per tubule, respectively (***Figure 2—figure supplement 2B and D***). However, in the absence of RAD1, tubules contained only 74.9 ± 7.5 TRA98-positive cells in 4-week-old mice and 47.8 ± 8.3 in 12-week-old males.

*Stra8-Cre* expression occurs as cells are committing to undergo meiosis (***Sadate-Ngatchou et al., 2008***). We therefore anticipated that the apoptosis and germ cell loss observed in *Rad1* CKO mice were due to meiotic defects. To address the possibility of pre-meiotic defects in *Rad1* CKO mice, we assessed mice at 8 days postpartum (dpp), prior to meiotic entry. H&E staining, along with TUNEL and TRA98 staining of sections from both control and *Rad1* CKO mice, showed no significant differences between genotypes at 8 dpp (***Figure 2E–G***, ***Figure 2—figure supplement 2B and D***). To further confirm that RAD1 loss did not affect cells prior to meiotic entry, we stained sections for LIN28, a marker of spermatogonial stem cells (SSCs), which have not initiated meiosis (***Aeckerle et al., 2012***). As expected, no significant differences in LIN28 staining were observed between genotypes in testes from mice at 8 dpp or 4 weeks of age (***Figure 2—figure supplement 2C and E***), consistent with the notion that RAD1 targeting is specific to meiotic cells. However, 12-week-old *Rad1* CKO mice had a significant decrease in LIN28-positive cells when compared to control mice. This later loss of LIN28-positive cells in *Rad1* CKO mice can be attributed to large-scale germ cell loss, which could indirectly disrupt the environment required for proper SSC proliferation and survival.

Next, staining of surface spread spermatocyte nuclei was performed to test how localization of 9-1-1 subunits was affected by RAD1 loss. SYCP3, a component of the SC, was used to visualize the five substages of prophase I. Consistent with prior results (***Freire et al., 1998***; ***Lyndaker et al., 2013a***),

RAD1 localized in control meiocytes during leptonema as foci (209.3 ± 23.9 RAD1 foci), including on chromosome cores that were not yet synapsed, and during zygonema on both unsynapsed and synapsed chromosome cores (208.2 ± 9.2 RAD1 foci) (*Figure 3A*). In mid-pachynema, RAD1 was present on fully synapsed core axes of autosomes as well as along the X and Y chromosomes (120.8 ± 27.3 RAD1 foci). By late-pachynema, RAD1 was no longer present on autosomes but continued to be abundant along the X-Y cores. RAD1 localization was completely absent in 43% of spermatocytes from 12-week-old *Rad1* CKO mice, whereas 100% of control cells showed proper RAD1 localization and abundance in zygotene- and pachytene-stage cells. The RAD1 localization observed in some *Rad1* CKO meiocytes could be attributed to cells that failed to undergo CRE-mediated recombination or in which RAD1 levels were not yet fully depleted. Consistent with the latter possibility, pachytene-stage *Rad1* CKO cells with detectable RAD1 focus formation had significantly fewer RAD1 foci than stage-matched control cells (106.0 ± 25.8 vs. 120.8 ± 27.3 RAD1 foci; p-value ≤ 0.0002; *Figure 3— figure supplement 1A*). The fact that *Rad1* CKO cells were prone to apoptosis as described below would be expected to eliminate cells lacking RAD1, leaving RAD1-intact meiocytes enriched among the remaining cells. Additional functional analyses (see below) indicated that, overall, approximately 72% of *Rad1* CKO cells had a meiotic defect (summarized in *Figure 3—figure supplement 1B*). We next evaluated how RAD1 disruption impacted RAD9A/B localization. In control samples, RAD9A and RAD9B localized to unsynapsed chromosomes as foci in leptotene-stage cells (145.5 ± 20.8 RAD9A foci; 211.5 ± 50.8 RAD9B foci) and to synapsed and unsynapsed chromosome cores in zygotene-stage cells (125.2 ± 25.6 RAD9A foci; 230.5 ± 40.5 RAD9B foci) (*Figure 3B and C*). By pachynema, RAD9A and RAD9B localized primarily as foci on autosomes and sex chromosome cores (107.3 ± 32.6 RAD9A foci; 125.5 ± 19.2 RAD9B foci). RAD9A and RAD9B foci were absent in 79 and 72% of *Rad1* CKO meiotic spreads, respectively.

*Rad1* CKO mice had no epididymal sperm (*Table 1*). To assess if *Rad1* CKO mice were infertile, we bred control and *Rad1* CKO mice with wild-type females. Control mice bred with wild-type females yielded 10 pregnancies and 66 viable pups, whereas *Rad1* CKO mice had no viable offspring from 15 matings with wild-type females. Overall, these results indicate that RAD1 disruption severely compromised spermatogenesis and fertility. Moreover, the reduced testis weight and increased apoptosis in *Rad1* CKO mice were more severe than those in mice with *Hus1* or *Rad9a* loss (*Lyndaker et al., 2013a*; *Vasileva et al., 2013*), suggesting a broader role for RAD1 in meiocytes.

## *Rad1* loss results in synapsis defects and increased DNA damage

During meiosis, SC formation is critical for homologous chromosomes to pair and fully synapse (*Zickler and Kleckner, 2015*). Co-staining for the SC markers SYCP1 and SYCP3 revealed that 59.5% ± 4.3% of meiocytes from *Rad1* CKO mice had whole chromosomes that remained unsynapsed and/or aberrant synapsis events involving multiple chromosomes, whereas 100% of meiocytes from control mice displayed normal homolog synapsis (*Figure 3D*, *Figure 3—figure supplement 2A*). RAD1 staining in meiocytes from 12-week-old *Rad1* CKO mice revealed that all cells that lacked RAD1 displayed abnormal synapsis, with an average of only eight fully synapsed chromosomes per cell (*Figure 3— figure supplement 2B*). Cells with asynapsis that contained four or more synapsed homologous chromosomes were classified as pachytene-like cells. Unless otherwise noted, subsequent analyses described below focused on this population of RAD1-deficient meiocytes.

The γH2AX staining pattern was similar in *Rad1* CKO and control spreads at leptonema and zygonema (*Figure 3E*). However, 97% of pachytene-like *Rad1* CKO cells showed γH2AX present at asynaptic sites, with no clear presence of a sex body (n = 98 cells, three CKO mice). Interestingly, a subset of asynaptic regions in *Rad1* CKO cells lacked detectable γH2AX staining (*Figure 3E*, white arrowheads; *Figure 3—figure supplement 2C*). Even in *Rad1* CKO spermatocytes with apparently normal synapsis, 15.1% ± 11.5% of cells exhibited defects in γH2AX staining on the XY body, with partial or no coverage of γH2AX on the Y chromosome or expansion of the γH2AX domain to encompass an autosome (n = 205 cells, four CKO mice; *Figure 3—figure supplement 2C*). By contrast, such γH2AX staining defects were observed in only 3.2% ± 1.5% of pachytene-stage control cells. Together, these data suggest that RAD1 loss perturbed DNA damage signaling at asynaptic sites and the XY body.

Given that spermatocytes from *Rad1* CKO mice exhibited significantly increased asynapsis, we next assessed meiotic progression in these cells by staining for the histone variant H1T and the

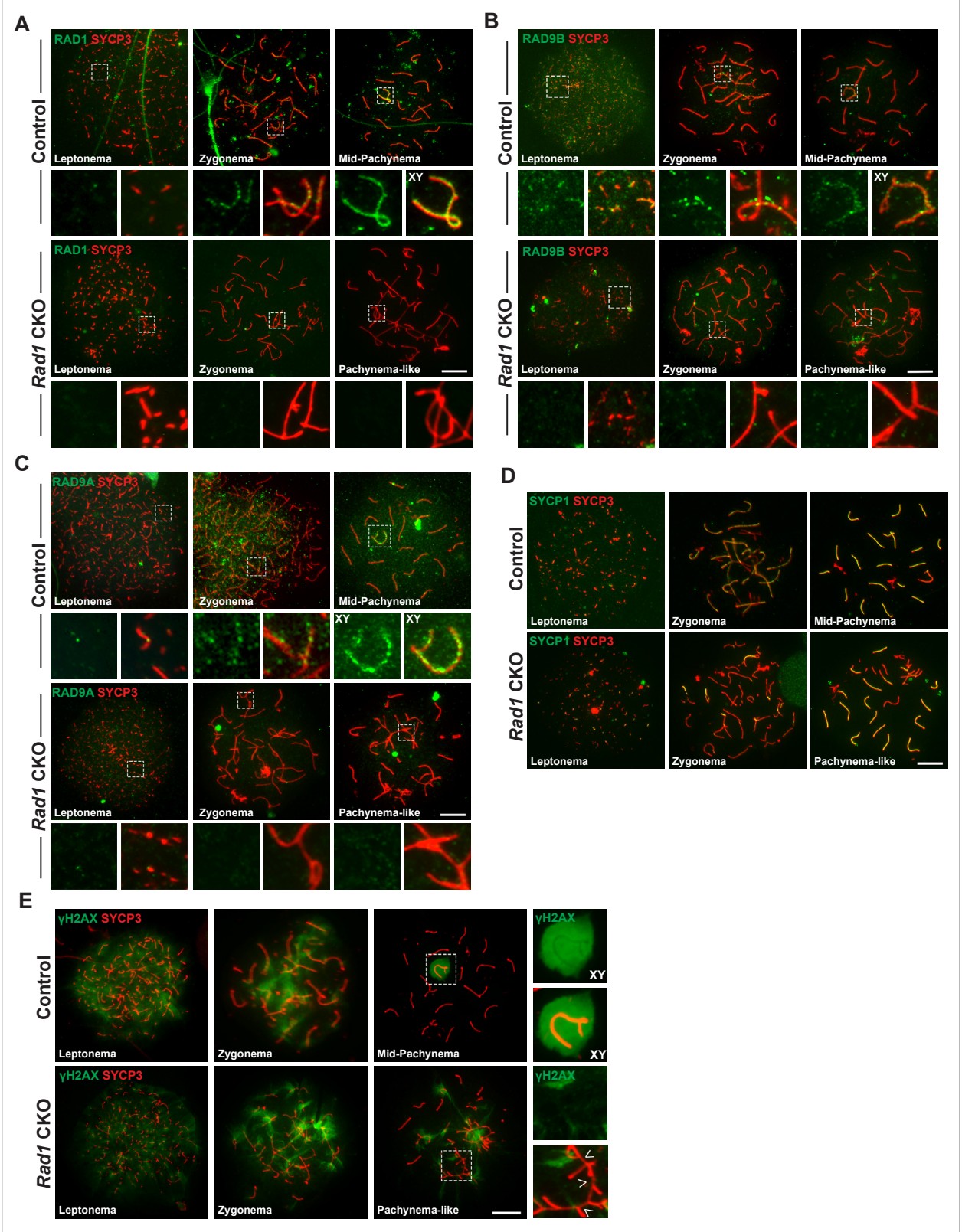

**Figure 3.** Testis-specific RAD1 loss disrupts 9-1-1 complex localization and causes defects in homolog synapsis and DNA damage signaling. (A-C) Meiotic spreads from 12-week-old control and *Rad1* conditional knockout (CKO) mice stained for RAD1 (**A**), RAD9B (**B**), or RAD9A (**C**). (**D**) Co-staining for SYCP1 and SYCP3 in meiotic spreads from 12-week-old control and *Rad1* CKO mice (three control mice, n = 156 cells; three CKO mice, n = 131 cells). *Rad1* CKO meiocytes with four or more synapsed chromosomes were categorized as pachytene-like. (**E**) γH2AX staining of meiotic spreads from control

*Figure 3 continued on next page*

*Figure 3 continued*

and *Rad1* CKO mice. Arrowheads in *Rad1* CKO spreads highlight regions of asynapsis lacking γH2AX staining (three control mice, n = 127 cells; five CKO mice, n = 205 cells). p-Values were calculated using Welch's unpaired *t*-test using GraphPad. Scale bar for A-E 10μm.

The online version of this article includes the following source data and figure supplement(s) for figure 3:

**Source data 1.** RAD1 foci counts in control and *Rad1* conditional knockout (CKO) spermatocytes.

**Source data 2.** Quantification of synapsed chromosomes in control and *Rad1* conditional knockout (CKO) spermatocytes.

**Source data 3.** Total MLH1 foci in control and *Rad1* conditional knockout (CKO) spermatocytes.

**Figure supplement 1.** *Rad1* conditional knockout (CKO) spermatocytes vary in the extent of RAD1 loss and meiotic defects.

**Figure supplement 2.** RAD1-deficient spermatocytes have synapsis defects and do not progress to mid-pachynema.

recombination marker MLH1. First, we questioned whether RAD1-deficient cells were able to progress past mid-pachynema. Histone variant H1T is a marker of mid-pachynema and later stage wild-type spermatocytes (*Cobb et al., 1999*). Control cells demonstrate H1T staining as they progress into mid-pachynema (*Figure 3—figure supplement 2D*). However, H1T staining was absent in *Rad1* CKO meiocytes with asynapsis, indicating that the cells failed to progress past mid-pachynema. By mid-pachynema, crossover sites are normally marked by MLH1 (*Eaker et al., 2002*). In contrast to the apparently normal MLH1 focus formation reported for *Hus1* CKO cells lacking the canonical 9A-1-1 complex (*Lyndaker et al., 2013a*), MLH1 foci were not detected in any *Rad1* CKO cells at the pachytene-like stage (*Figure 3—figure supplement 2E*), further suggesting that *Rad1* CKO meiocytes with asynapsis fail to progress beyond early/mid-pachynema. Together, the observations of γH2AX abnormalities and SC defects in *Rad1* CKO cells indicate important roles for 9-1-1 complexes in ensuring homologous chromosome synapsis and appropriate DDR signaling in response to asynapsis.

## DSB repair is compromised in *Rad1* CKO spermatocytes

Localization of canonical 9A-1-1 subunit RAD9A to chromatin cores requires SPO11-induced DSBs (*Lyndaker et al., 2013a*), and testis-specific *Hus1* or *Rad9a* CKO results in persistent meiotic DSBs with delayed repair kinetics (*Lyndaker et al., 2013a*; *Vasileva et al., 2013*). We therefore investigated how RAD1 loss impacts DSB repair. Following MRE11-RAD50-NBS1 (MRN)-mediated resection of SPO11-induced meiotic DSB, meiosis-specific with OB domains (MEIOB) and RPA localize to the ssDNA overhangs prior to RAD51 and DMC1 loading (*Hinch et al., 2020*; *Luo et al., 2013*; *Shi et al., 2019*). In control spermatocytes, RPA and MEIOB foci are abundant in early prophase I and diminish as DSBs are repaired (*Figure 4A–D*). RPA and RAD1 both formed foci on meiotic chromosome cores, but the extent of co-localization was modest, consistent with the notion that RPA coats single-stranded DNA, whereas 9-1-1 is loaded at recessed DNA ends (*Figure 4—figure supplement 1A*). *Rad1* CKO testes had on average 50 fewer RPA foci than controls in leptotene-stage cells (194.4 ± 54.4 control; 145.8 ± 37.4 CKO; *Figure 4A and B*). Intriguingly, RPA foci in *Rad1* CKO cells appeared larger than those in control cells (*Figure 4A*, *Figure 4—figure supplement 1B*). In the absence of RAD1, MEIOB focus formation on chromatin cores in leptotene-stage cells was also significantly decreased as compared to control cells (230.4 ± 45.4 control; 125.8 ± 36.6 CKO) (*Figure 4C and D*). In control spermatocytes, MEIOB and RPA levels on meiotic chromosome cores decreased as the cells progressed into pachynema (115.1 ± 26.6 control MEIOB; 52.3 ± 40.5 control RPA), whereas *Rad1* CKO cells showed persistence of MEIOB and RPA staining (132.1 ± 38.0 CKO MEIOB; 100.0 ± 44.3 CKO RPA).

**Table 1.** Analysis of epididymal sperm counts and fertility in *Rad1* conditional knockout (CKO) and control mice.

| Genotype | No. males | Epididymal sperm count (×10⁶) | No. matings | No. copulatory plugs | No. pregnancies | Total viable pups |
|---|---|---|---|---|---|---|
| Control | 3 | 16.6 ± 4.5 | 12 | 12 | 10 | 66 |
| *Rad1* CKO | 3 | 0.0 ± 0 | 15 | 15 | 0 | 0 |

Male *Rad1* CKO mice at 8-12 weeks of age were bred to 6-week-old wild-type FVB female mice.

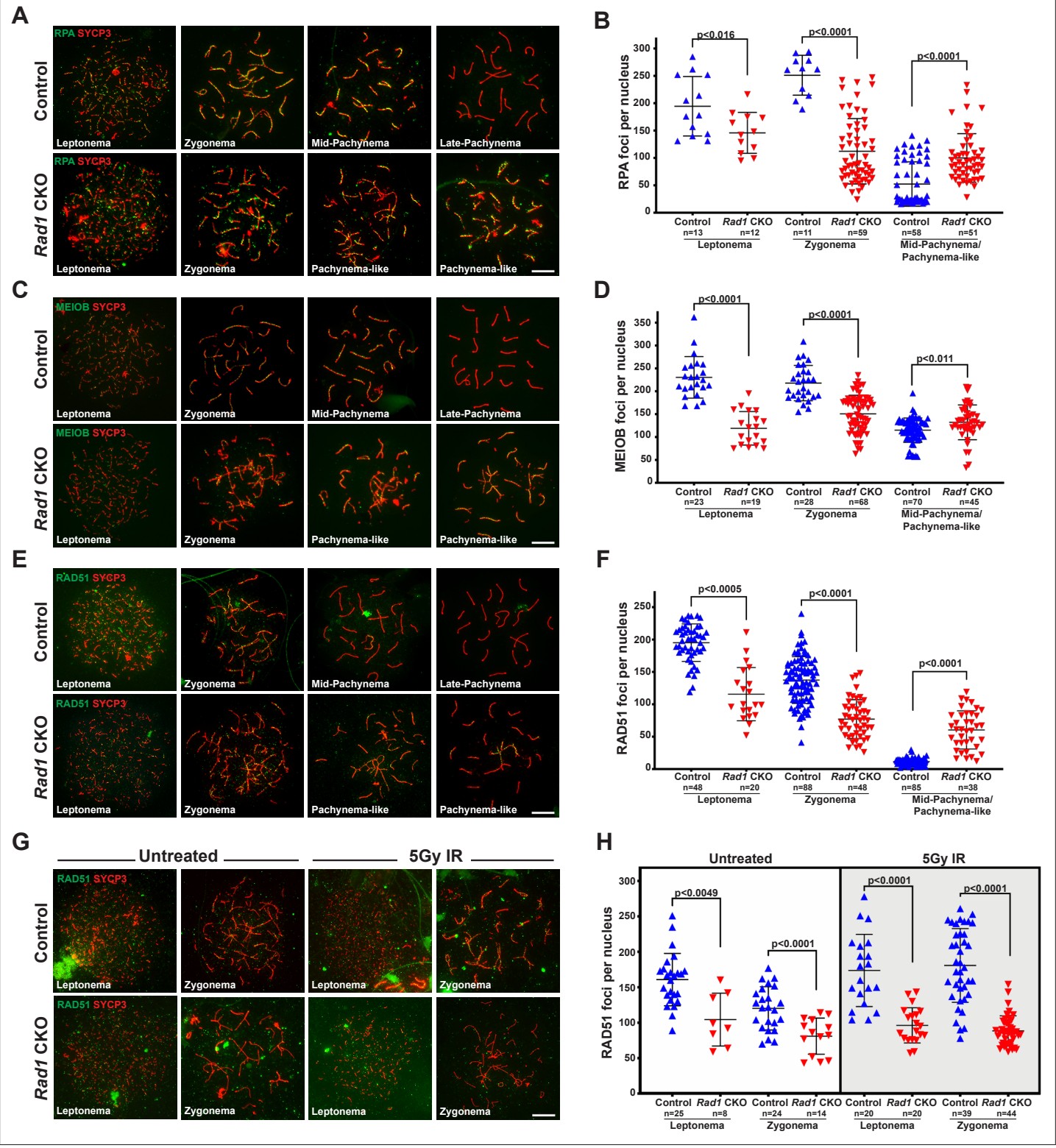

**Figure 4.** Double-strand break (DSB) repair is compromised in the absence of 9-1-1 complexes. (**A, B**) Representative images (**A**) and quantification (**B**) of RPA2 staining of meiotic spreads from 12-week-old control and *Rad1* conditional knockout (CKO) mice (three mice per genotype analyzed; n = total cells analyzed). (**C, D**) Representative meiotic spread images for ssDNA marker MEIOB (**C**) and quantifications (**D**) from 12-week-old control and *Rad1* CKO mice (three mice per genotype analyzed; n = total cells analyzed). (**E, F**) Representative meiotic spread images of RAD51 (**E**) and quantifications (**F**) from 12-week-old control and *Rad1* CKO mice (five control and six CKO mice; n = total cells analyzed). (**G, H**) 8-week-old control

*Figure 4 continued on next page*

*Figure 4 continued*

and *Rad1* CKO mice were irradiated with 5 Gy ionizing radiation (IR) and collected 1 hr post IR. Representative RAD51 meiotic spread images (**G**) and quantifications (**H**) (two control and two CKO mice; n = total cells analyzed). p-Values were calculated using Welch's unpaired *t*-test in GraphPad. Scale bars for A, C, E and G 10µm.

The online version of this article includes the following source data and figure supplement(s) for figure 4:

**Source data 1.** Total foci counts for RPA, MEIOB, and RAD51 in spermatocytes from control and *Rad1* conditional knockout (CKO) mice, as well as RAD51 and RPA in spermatocytes from irradiated control and *Rad1* CKO mice.

**Figure supplement 1.** The 9-1-1 complexes are critical for proper localization of meiotic double-strand break (DSB) repair proteins.

During prophase I in wild-type spermatocytes, RAD51 and DMC1 displace MEIOB and RPA from the ssDNA overhangs and drive the subsequent steps of homology search and strand invasion (*Gray and Cohen, 2016*; *Hinch et al., 2020*). The persistence of MEIOB and RPA foci in *Rad1* CKO spermatocytes suggested that RAD1 loss might perturb RAD51 loading. On average, leptotene-stage cells from control mice contained 195.2 ± 29.0 RAD51 foci, whereas *Rad1* CKO cells at the same stage had 115.7 ± 41.1 RAD51 foci (*Figure 4E and F*). RAD51 foci continued to be significantly lower in zygotene-stage *Rad1* CKO meiocytes, which contained 77.1 ± 30.0 RAD51 foci per cells as compared to 137.5 ± 36.1 in controls. In control samples, RAD51 foci levels decreased as cells progressed from zygonema to pachynema, reflecting the successful repair of DSBs. However, *Rad1* CKO spermatocytes retained relatively high levels of RAD51 foci in pachytene-like-stage cells (60.4 ± 29.5 RAD51 foci) as compared to control pachytene-stage meiocytes (11.5 ± 4.9 RAD51 foci) (*Figure 4F*). These results for RAD51 localization in *Rad1* CKO spermatocytes differed from those in *Hus1* CKO mice, where RAD51 appeared normal in early prophase and then was aberrantly retained at a small number of sites in pachytene-stage cells (*Lyndaker et al., 2013a*). Together, these results suggest that the 9-1-1 complexes are critical for DSB processing and repair during mammalian meiosis and that absence of RAD1, or to a lesser extent HUS1, leaves persistent unrepaired DSBs.

The abnormal localization profiles for MEIOB, RPA, and RAD51 observed in *Rad1*-deficient spermatocytes raised the possibility that DSB formation was impaired. To determine whether the defects were related to DSB formation or the subsequent repair steps, we treated *Rad1* CKO and control mice with 5 Gy ionizing radiation (IR), harvested testes 1 hr post treatment, and quantified RPA and RAD51 focus formation in leptotene- and zygotene-stage cells. Since exogenously induced DSBs are repaired via meiotic processes in early stages of prophase I (*Enguita-Marruedo et al., 2019*), this approach allowed us to test whether the alterations in DSB markers in *Rad1* CKO cells were due to reduced DSB formation or a DSB repair defect. Upon DSB induction via irradiation, control mice showed increased RPA and RAD51 focus formation at early prophase I stages (*Figure 4G and H*, *Figure 4—figure supplement 1C and D*). By contrast, irradiation did not induce increased focus formation by RPA or RAD51 in *Rad1* CKO spermatocytes. Together, these results highlight the importance of the 9-1-1 complexes for meiotic DSB repair.

## RAD1 deficiency compromises meiotic ATR signaling

Given that the canonical 9A-1-1 complex plays a central role in stimulating ATR kinase activity in somatic cells, we sought to determine the effect of RAD1 loss on the localization of ATR and its substrates in meiocytes. ATR localizes to unsynapsed regions at early stages of prophase I, and by pachynema it is sequestered mainly at the XY body where it initiates MSCI (*Abe et al., 2020*; *Turner, 2015*). Cells from *Rad1* CKO mice with synapsis defects showed ATR localization only at a subset of asynaptic regions (*Figure 5A*).

TOPBP1 is required for ATR activation following replication stress (*Blackford and Jackson, 2017*) and interacts with ATR during meiosis to ensure that meiotic silencing is properly initiated (*Ellnati et al., 2017*; *Jeon et al., 2019*). In control meiocytes, TOPBP1 was observed as discrete foci on unsynapsed chromosome cores throughout leptonema and zygonema (*Figure 5B*). At pachynema, TOPBP1 was found exclusively along the unsynapsed regions of the X and Y and present as a faint cloud on XY chromosome loops. By contrast, in pachytene-like stage *Rad1* CKO cells, TOPBP1 localized to only a subset of asynaptic sites, failing to coat the entirety of unsynapsed chromosome cores, similar to the pattern observed for ATR. These findings suggest a role for the 9-1-1 complexes in promoting ATR and TOPBP1 localization to unsynapsed chromatin, although this is, at least in part, likely an indirect

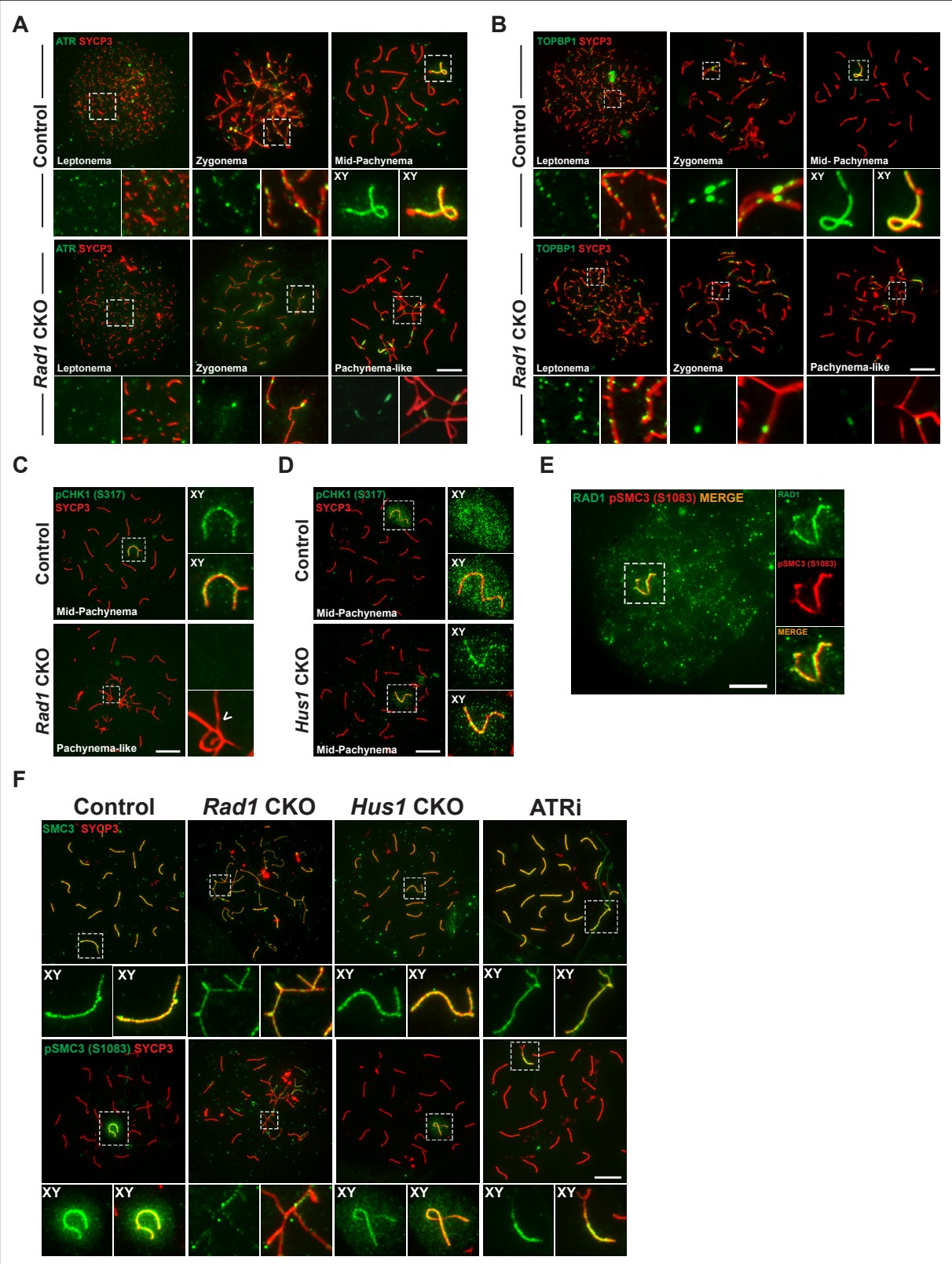

**Figure 5.** Key ATR phosphorylation events for double-strand break (DSB) repair and cohesion are dependent upon 9-1-1 complexes. (**A**) ATR localization in meiotic spreads from control and *Rad1* conditional knockout (CKO) 12-week-old mice (three control mice, n = 171 cells; three CKO mice, n = 146). (**B**) Representative images of TOPBP1 localization in meiotic spreads from 12-week-old control and *Rad1* CKO mice (three control mice, n = 130 cells; three CKO mice, n = 129). (**C, D**) Representative images of phospho-CHK1 (S317) localization in *Rad1* CKO (**C**) and *Hus1* CKO mice (**D**) (*Rad1* CKO:

*Figure 5 continued on next page*

*Figure 5 continued*

three control mice, n = 125 cells; three CKO mice, n = 120 cells; *Hus1* CKO: two control mice, n = 107 cells; three CKO mice, n = 191 cells). Arrowhead indicates a region of asynapsis. (**E**) Co-staining of RAD1 and pSMC3 (S1083) in wild-type spermatocytes. (**F**) Representative images of SMC3 and pSMC3 (S1083) localization in pachytene and pachytene-like cells from control, *Rad1* CKO, *Hus1* CKO, and ATRi-treated mice. Scale bars 10µm.

The online version of this article includes the following source data and figure supplement(s) for figure 5:

**Source data 1.** Evolutionary rate covariation (ERC) calculations for 9-1-1 complex subunits and meiosis I-related proteins.

**Figure supplement 1.** Phosphorylation of CHK1 and SMC3 is reduced in the absence of 9-1-1 complexes.

**Figure supplement 2.** Evolutionary rate covariation (ERC) network of meiosis I proteins.

effect of the extensive asynapsis in *Rad1* CKO cells as the available pool of silencing factors can be insufficient to localize to all asynaptic sites under such circumstances (*Mahadevaiah et al., 2008*).

The best characterized ATR substrate in somatic cells is the transducer kinase CHK1. CHK1 has been proposed to play a role in meiotic DSB repair and is suggested to aid progression through prophase I by removal of DDR proteins such as γH2AX from autosomes (*Abe et al., 2018*; *Fedoriw et al., 2015*; *Pacheco et al., 2018*). In wild-type cells, CHK1 phosphorylation (S317) occurs during leptonema and zygonema at unsynapsed chromosomes. During pachynema, pCHK1 (S317) is apparent on XY cores and as a cloud over the sex body (*Figure 5C*). Interestingly, in the *Rad1* CKO mutant, pCHK1 was absent at all stages of prophase I. Reduced CHK1 phosphorylation at ATR target sites S317 and S345 in the absence of RAD1 was confirmed by whole testis immunoblotting (*Figure 5—figure supplement 1A*). By contrast, meiotic spreads from *Hus1* CKO mice showed normal patterns of CHK1 (S317) phosphorylation (*Figure 5D*). That meiotic CHK1 phosphorylation is normal in the absence of HUS1 but disrupted by RAD1 loss suggests that alternative 9-1-1 complexes play an important role in activating the transducer kinase CHK1 during meiotic prophase I.

The cohesin subunit SMC3 has been implicated as another likely meiotic ATR substrate (*Fukuda et al., 2012*). Loss of meiosis-specific cohesins results in phenotypes similar to those in *Rad1* CKO mice, including SC assembly defects, impaired synapsis, and DSB repair failure and synapsis defects (*Challa et al., 2019*; *Eijpe et al., 2003*; *Ishiguro, 2019*; *Llano et al., 2012*; *Ward et al., 2016*). Notably, phosphoproteomic analyses revealed that phosphorylation of cohesin complex (SMC3) and SC (SYCP1, SYCP2, and SYCP3) components during meiosis is RAD1- and ATR-dependent (*Sims et al., 2021*). Furthermore, correlated evolutionary relationships, as measured by ERC analysis, were observed between genes encoding 9-1-1 subunits and those encoding cohesin and SC proteins, including SMC1β, RAD21L1, SYCP2, and SYCE1 (*Figure 5—figure supplement 1B, C*). We therefore further interrogated the relationship between RAD1 and SMC3. RAD1 and pSMC3 (1083) co-localized at the XY in pachynema-stage wild-type spermatocytes (*Figure 5E*). In control meiocytes, SMC3 was observed on chromatin cores throughout prophase I and was phosphorylated specifically at unsynapsed chromatin cores during leptonema and zygonema, and at the unsynapsed regions of the XY in mid-pachynema (*Figure 5F*). Although total SMC3 loading was unaffected by RAD1 loss, *Rad1* CKO spermatocytes showed reduced accumulation of phosphorylated SMC3 (pSMC3 S1083) at unsynapsed chromatin regions in pachytene-like cells as compared to mid-pachytene-stage control cells. Western blot analysis of whole testis lysates confirmed that SMC3 phosphorylation (pSMC3 S1083) was significantly reduced in testes from *Rad1* CKO mice (*Figure 5—figure supplement 1D and E*). Unlike *Rad1* CKO spermatocytes, *Hus1* CKO cells had grossly normal pSMC3 (S1083) localization to the XY in pachytene-stage spermatocytes (*Figure 5F*). To determine if SMC3 (1083) phosphorylation was ATR dependent, we performed immunostaining of spermatocytes from wild-type C57BL/6 male mice treated with the ATR inhibitor AZ20 (ATRi). Similar to the effects of RAD1 loss, acute ATRi treatment caused a decrease in pSMC3 (S1083) at X and Y chromatin loops and cores despite the fact that SMC3 localization to chromosome cores appeared normal, suggesting a specific defect in SMC3 phosphorylation. Together, these results suggest that 9-1-1 complexes and ATR act in conjunction to regulate meiotic cohesin phosphorylation.

## Loss of 9-1-1 complexes disrupts ATR-mediated meiotic silencing

Given that ATR is a primary regulator of MSCI (*Fedoriw et al., 2015*; *Pacheco et al., 2018*; *Turner et al., 2005*; *Widger et al., 2018*), the defects in ATR signaling noted above prompted us to assess meiotic silencing in *Rad1* CKO mice. We first looked upstream of ATR and examined whether the

absence of RAD1 impacted localization of HORMA-domain proteins 1 and 2 (HORMAD1 and HORMAD2). The presence of HORMADs at unsynapsed chromatin is important for meiotic silencing, and HORMAD1 is required for ATR recruitment to unsynapsed sites (*Fukuda et al., 2012*; *Kogo et al., 2012*; *Shin et al., 2010*; *Wojtasz et al., 2012*). In control cells, HORMAD1 and HORMAD2 were observed during early prophase I at chromosomal regions that were not yet synapsed (*Figure 6A and B*). By mid-pachynema, the HORMADs localized strictly at the unsynapsed regions of the XY, similar to the localization of ATR. Notably, RAD1 loss did not alter HORMAD1 or HORMAD2 localization to unsynapsed regions. Furthermore, the HORMADs were observed to entirely coat unsynapsed chromosome regions in *Rad1* CKO cells, in contrast to the failure of ATR and TOPBP1 to localize to all unsynapsed sites.

ATR phosphorylates HORMAD1 (S375) and HORMAD2 (S271) at asynaptic regions (*Fukuda et al., 2012*; *Royo et al., 2013*). In control cells, HORMAD2 (S271) phosphorylation was observed on the X and Y chromosome cores in mid-pachytene-stage cells as expected (*Figure 6C*). However, in pachytene-like *Rad1* CKO cells, phosphorylated HORMAD2 was detected at only a subset of unsynapsed regions. That HORMAD2 localized properly in the absence of RAD1 but lacked phosphorylation at an ATR-regulated site further supports the notion that meiotic ATR signaling requires the 9-1-1 complexes. BRCA1 is another key meiotic silencing factor, and its localization is interdependent with that of ATR in a stage-specific manner, with conditional *Atr* deletion disrupting BRCA1 localization to XY axial elements (*Mahadevaiah et al., 2008*; *Royo et al., 2013*). BRCA1 failed to localize properly in *Rad1* CKO cells with extensive asynapsis, coating only a subset of asynaptic regions much like what was observed for ATR and TOPBP1 in RAD1-deficient cells (*Figure 6D*).

The defects in ATR signaling observed in *Rad1* CKO mice suggested that disruption of 9-1-1 complexes might impair meiotic silencing. To test this possibility, we first examined the localization of RNA Pol II, the exclusion of which from the XY body is an indicator of MSCI (*Figure 6E*). In control cells, RNA Pol II was excluded from the sex body in the vast majority of cells, with only 8.8% ± 2.4% of cells showing any RNA Pol II at the XY chromosomes. In *Rad1* CKO meiocytes with extensive asynapsis, RNA Pol II was diffusively distributed, but the sex chromosomes cannot be distinguished in such cases. We next assessed RNA Pol II in *Rad1* CKO cells with apparently normal synapsis and observed that 27.8% ± 19.1% of such cells failed to fully exclude RNA Pol II from the XY body. This contrasted with our previous analysis of *Hus1* CKO spermatocytes with disruption of the canonical 9A-1-1 complex, in which RNA Pol II was properly excluded from the sex body (*Lyndaker et al., 2013a*). Finally, we directly evaluated meiotic silencing via RNA fluorescent in situ hybridization (FISH) for the X-chromosome gene *Scml2* that should be silenced in early pachynema-stage cells (*Royo et al., 2010*). Because autosome asynapsis antagonizes MSCI (*Mahadevaiah et al., 2008*), the analysis of *Scml2* expression focused on cells with normal homolog synapsis and excluded those with asynapsis. Inappropriate *Scml2* expression was detected in 7.1% ± 0.6% of early pachytene-stage control cells, but 28.9% ± 3.2% of *Rad1* CKO cells (p<0.0001; *Figure 6F*), indicating that meiotic silencing was disrupted by RAD1 loss. This quantification underestimates the extent of the silencing defect upon RAD1 loss since some cells with apparently normally synapsis in *Rad1* CKO mice retain normal RAD1 expression. Nevertheless, approximately 30% of *Rad1* CKO cells with apparently normal synapsis show both defective exclusion of RNA Pol II from the XY body as well as inappropriate expression of an X-linked gene. Together with the clear evidence for defective ATR signaling upon RAD1 loss, these results demonstrate the importance of the canonical and alternative 9-1-1 complexes in promoting ATR-mediated MSCI.

## Discussion

Here, we report that testis-specific RAD1 loss results in homolog asynapsis, compromised DSB repair, faulty ATR signaling, and impaired meiotic silencing (*Figure 6G*). Previous analyses of the canonical 9A-1-1 complex in meiosis revealed that loss of *Hus1* or *Rad9a* leads to a small number of unrepaired DSBs that trigger germ cell death (*Lyndaker et al., 2013a*; *Vasileva et al., 2013*). Yet, homolog synapsis, ATR activation, and meiotic silencing all are grossly normal in the absence of the canonical 9A-1-1 subunits HUS1 and RAD9A. The expanded roles for RAD1 identified here are consistent with its ability to additionally interact with RAD9B and HUS1B, paralogs that evolved in higher organisms and are highly expressed in germ cells. The dependency of RAD9A and RAD9B localization as well as meiotic ATR activation on RAD1 supports the idea that RAD1-containing alternative 9-1-1 complexes

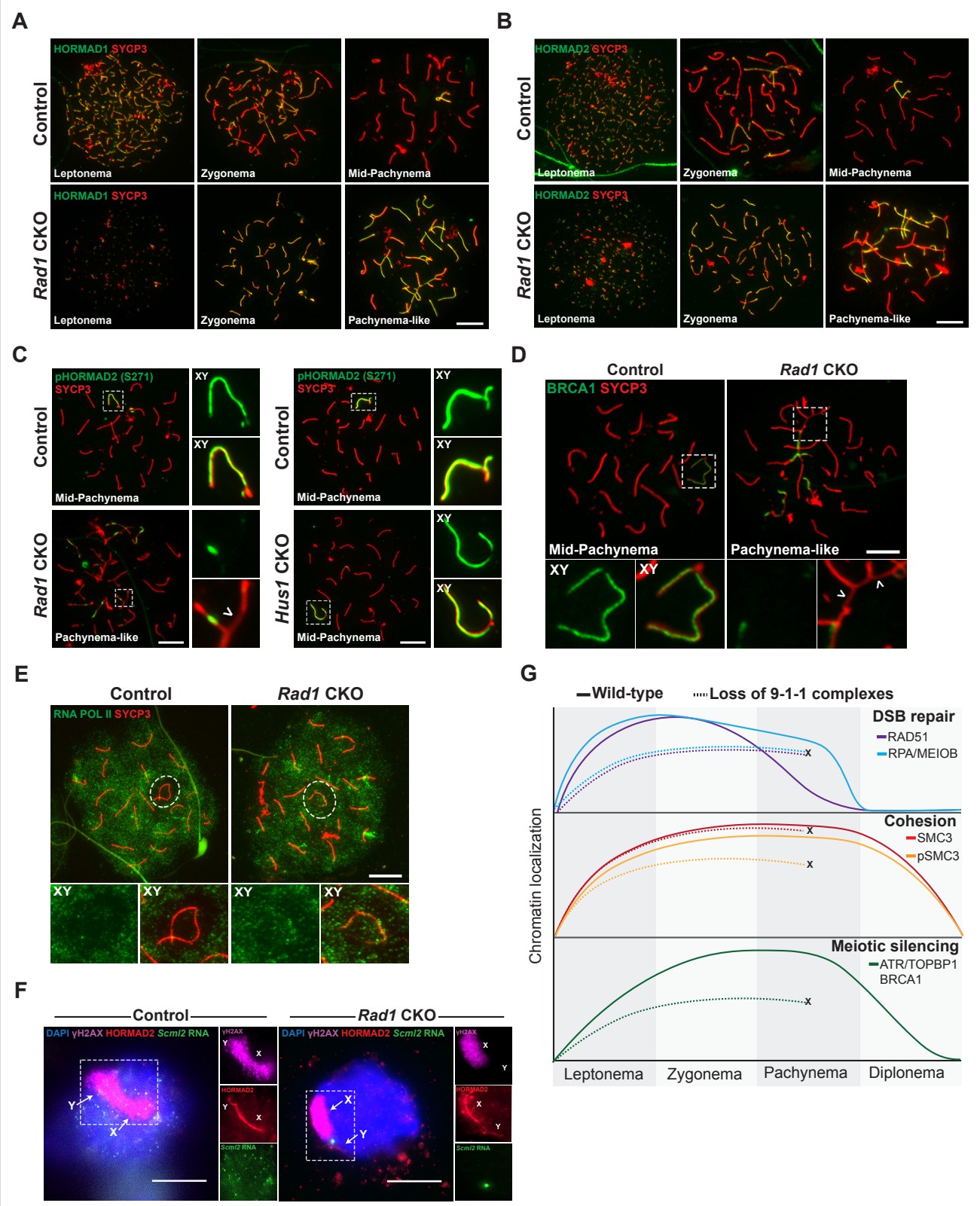

**Figure 6.** 9-1-1 complexes are required for ATR-mediated meiotic sex chromosome inactivation. (**A, B**) Representative images of HORMAD1 (**A**) and HORMAD2 (**B**) localization in meiotic spreads from 12-week-old control and *Rad1* conditional knockout (CKO) mice (three control mice, n = 146 cells; three CKO mice, n = 119 cells). (**C**) Representative images of phospho-HORMAD2 (S271) localization in meiotic spreads from 12-week-old control and *Rad1* CKO mice (*Rad1* CKO: three control mice, n = 178 cells; three CKO mice, n = 146 cells; *Hus1* CKO: two control mice, n = 189 cells; three CKO

Figure 6 continued

mice, n = 145 cells). Arrowhead indicates a region of asynapsis. (**D**) Representative images of BRCA1 localization in meiotic spreads from 12-week-old control and *Rad1* CKO mice (two control, n = 110 cells; two CKO, n = 149 cells). Arrowheads indicate regions of asynapsis. (**E**) Representative images of RNA Pol II staining in meiotic spreads from 12-week-old control and *Rad1* CKO mice (three control, n = 125 cells; three CKO, n = 98 cells). Note that the *Rad1* CKO cell has apparently normal synapsis but incomplete exclusion of RNA Pol II from the sex body. (**F**) RNA fluorescent in situ hybridization for *Scml2* in fully synapsed, pachytene-stage control and *Rad1* CKO cells, with co-staining for γH2AX and HORMAD2 (three control mice, n = 29 cells; three CKO mice, n = 45 cells). (**G**) Summary graphic depicting the localization of key meiotic factors in wild-type versus *Rad1* CKO spermatocytes. Loss of 9-1-1 complexes resulted in failure to progress to late pachytene, as depicted by the 'X.' Double-strand break (DSB) repair markers, such as RAD51, were reduced in the absence of the 9-1-1 complexes. The cohesin subunit SMC3 localized properly in the absence of 9-1-1 subunits, but its phosphorylation was impaired in *Rad1* CKO spermatocytes. Meiotic silencing factors such as ATR, TOPBP1, and BRCA1 also failed to localize properly in the absence of the 9-1-1 complexes. Scale bars for A-F 10μm.

(9B-1-1 and 9B-1-1B complexes) mediate essential roles in meiotic DSB repair, homolog synapsis, and MSCI, although we cannot exclude the possibility that RAD1 also functions independently of these heterotrimeric complexes.

In *Rad1* CKO spermatocytes, RAD51 loading onto meiotic chromosome cores was significantly reduced at leptonema and zygonema relative to controls. By mid-pachynema in control cells, RAD51 chromatin levels are low as DSB repair is concluding, but substantial RAD51 focus formation was still observed in pachytene-like *Rad1* CKO cells, suggesting major DSB repair defects. The meiotic DSB repair defects following RAD1 loss are similar to those previously observed in *Atr* loss-of-function mouse models. Zygotene-stage cells from a Seckel mouse model with disrupted ATR expression have decreased RAD51 and DMC1 loading (*Pacheco et al., 2018*), similar to that of spermatocytes lacking RAD1. Meiotic RAD51 focus formation did not increase further in *Rad1* CKO meiocytes after irradiation. These findings suggest that, similar to what is observed in *Atr*-defective spermatocytes (*Pacheco et al., 2018*; *Widger et al., 2018*), the defects in RAD51 loading were not due to decreased numbers of SPO11-induced DSBs in *Rad1* CKO mice, highlighting an important role for the 9-1-1 complexes in the subsequent repair of meiotic DSBs.

Unlike what is observed in *Atr* mutants and ATR inhibitor-treated mice, localization of ssDNA markers MEIOB and RPA to meiotic cores was significantly reduced in the absence of RAD1. The 9-1-1 complex is well established to modulate DNA end resection, having stimulatory or inhibitory effects in different contexts. In both yeast and mammals, the resection-stimulatory effects of the 9-1-1 complex involve recruitment of the Exo1 and Dna2 nucleases to DNA (*Blaikley et al., 2014*; *Karras et al., 2013*; *Ngo et al., 2014*; *Ngo and Lydall, 2015*). Phosphoproteomic analysis of *Rad1* CKO testes and ATRi-treated mice also revealed a significant decrease in phosphorylation of proteins involved in DNA end resection, including RAD50, NBS1, and CTIP (*Sims et al., 2021*). Conditional *Nbs1* knockout in testes was previously reported to cause a decrease in chromatin loading of RPA, MEIOB, and RAD51 (*Zhang et al., 2020*), similar to that in *Rad1* CKO mice, further suggesting potential functional interplay between the 9-1-1/ATR signaling axis and MRN complex during meiosis.

Somatic ATR activation via 9A-1-1/TOPBP1 interaction is well established; however, ATR and TOPBP1 localization in spermatocytes was unperturbed in the absence of *Hus1*. ATR-dependent processes such as sex body formation and meiotic silencing still occurred without HUS1 or RAD9A (*Lyndaker et al., 2013a*; *Vasileva et al., 2013*). By contrast, the localization of ATR, TOPBP1, and BRCA1 to unsynapsed regions was compromised in *Rad1* CKO spermatocytes. ATR and BRCA1 work in a positive feedback loop to encourage meiotic silencing (*Royo et al., 2013*; *Turner et al., 2004*), and the canonical and alternative 9-1-1 complexes may also be part of this regulatory circuitry. Phosphorylation of some ATR targets, such as H2AX and HORMAD2, still occurred in *Rad1* CKO spermatocytes but only at a subset of unsynapsed chromatin regions. It should be noted that HORMAD1 and HORMAD2 localized appropriately to all unsynapsed regions independently of RAD1, indicating that HORMAD localization was not sufficient to drive ATR signaling and highlighting essential roles for the 9-1-1 complexes in meiotic ATR activation, likely through interaction with TOPBP1. Other ATR substrates were more profoundly affected by RAD1 loss. CHK1 phosphorylation during meiosis was absent in *Rad1* CKO mice but present in *Hus1* CKO mice, suggesting that HUS1-independent alternative 9-1-1 complexes are necessary for meiotic CHK1 activation. CHK1 regulates the timing of both removal of γH2AX from autosomes and establishment of an ordered γH2AX domain at the sex body, but is not essential for MSCI (*Abe et al., 2018*). Proper loading of RAD51 and DMC1 onto chromatin also depends on CHK1 and ATR function (*Pacheco et al., 2018*). Thus, the CHK1 phosphorylation

defects described here when all 9-1-1 complexes are disrupted could contribute to multiple pheno-types observed in *Rad1* CKO spermatocytes, particularly faulty DSB repair.

Reduced SC protein phosphorylation also is observed in *Rad1* CKO and ATRi-treated mice (**Sims et al., 2021**), suggesting a role for the 9-1-1 complexes in mammalian SC formation. Studies in *Saccharomyces cerevisiae* show that direct interaction between the 9-1-1 complex and an SC component, Red1, is required for both meiotic checkpoint signaling and SC formation (**Eichinger and Jentsch, 2010**). Additionally, the budding yeast 9-1-1 complex also directly interacts with Zip3, a member of the ZMM (Zip, Mer, Msh) group of proteins that promote initiation of SC formation and crossover recombination. Notably, budding yeast 9-1-1 and clamp loader mutants show reduced ZMM assembly on chromosomes, impaired SC formation, and reduced interhomolog recombination (**Eichinger and Jentsch, 2010**; **Ho and Burgess, 2011**; **Shinohara et al., 2019**; **Shinohara et al., 2015**).

ATR has been linked to the phosphorylation of the cohesion complex component SMC3 (**Fukuda et al., 2012**), and SMC3 phosphorylation at a canonical ATR S/T-Q motif (S787) is downregulated in both *Rad1* CKO and ATRi-treated mice (**Sims et al., 2021**). Our analyses found SMC3 localization to meiotic chromosome cores to be unperturbed, but SMC3 phosphorylation at S1083 was dependent on RAD1 and ATR. NIPBL, which functions in association with Mau2 as an SMC loader that localizes to chromosomal axes from zygonema to mid-pachynema (**Visnes et al., 2014**) also had reduced phosphorylation in testes from *Rad1* CKO and ATRi-treated mice (**Sims et al., 2021**). Interestingly, in *Caenorhabditis elegans,* SCC-2[NIPBL] loss disrupts DSB processing, cohesin loading, and 9-1-1 recruitment to DNA damage sites (**Lightfoot et al., 2011**).

We also used ERC analysis to reveal potential mechanistic roles for 9-1-1 subunits. ERC analysis can infer functional protein partners based upon correlated rates of evolutionary change. ERC network analysis of the relationship between proteins involved in meiosis I and the 9-1-1 subunits revealed a clustering of RAD9B, RAD1, and HUS1, while RAD9A and HUS1B did not show high ERC values with the other 9-1-1 subunits and had mostly separate network interactions (**Figure 5—figure supplement 2A and B**). This approach also highlighted significant evolutionary correlations between the genes encoding 9-1-1 complex subunits and those encoding proteins involved in SC formation, such as SYCP1, SYCE1, SYCE1L, and SYCE2, in addition to RAD21, RAD21L, and SMC1β, which are involved in cohesion. Defects in homolog synapsis in *Rad1* CKO mice, together with the decreased cohesin phosphorylation, further implicate the 9-1-1 complexes in these key aspects of meiotic chromosome structure. However, further exploration of the mechanisms underlying the interactions between SC proteins, cohesin, and the 9-1-1 complexes is necessary and may provide insights into the basis for the DSB repair defects in *Rad1* CKO mice as proper SC formation and cohesin function is important for DSB repair (**Ishiguro, 2019**).

In mitotic cells, ATR activation is dependent on the 9A-1-1/TOPBP1 axis under cellular stress, while ATR activation during unperturbed conditions relies on ETAA1 (**Bass and Cortez, 2019**). The potential contributions of ETAA1 to meiotic ATR activation have yet to be directly assessed. However, *Etaa1* expression is low in germ cells and in spermatocytes in particular, and ETAA1 was reported to not localize to the XY chromosomes during meiosis (**Ellnati et al., 2017**). Mice expressing a ETAA1 mutant with a 42 amino acid deletion show signs of replication stress but are fertile (**Miosge et al., 2017**), further hinting at a predominant role for the 9-1-1/TOPBP1 axis as a primary regulator of meiotic ATR activation. Understanding the differential roles of 9-1-1/TOPBP1 and possibly ETAA1 in meiotic ATR activation may highlight different modes of structure-specific ATR activation that are coupled with distinct downstream outputs.

Although this study highlights key meiotic functions of both canonical and alternative 9-1-1 complexes, our approach does not resolve the relative importance of the DNA repair and checkpoint signaling roles of the 9-1-1 complexes during meiosis. Previous studies identified separable roles for 9-1-1 complexes in ATR activation via TOPBP1 interaction, and DNA repair protein scaffolding through the outer surface of 9-1-1 clamps (**Lim et al., 2015**). The loss of 9-1-1 complex formation and loading in *Rad1* CKO mice disrupts both of these roles. In budding yeast, the direct interactions between the 9-1-1 complex and Red1 as well as Zip3, together with additional evidence that the roles for 9-1-1 in SC formation and recombination can be distinguished from those of Mec1 (ATR), provide compelling support for the notion that the 9-1-1 complex executes signaling-independent functions during meiosis, aside from its roles in checkpoint signaling (**Eichinger and Jentsch, 2010**; **Shinohara et al., 2019**; **Shinohara et al., 2015**). In the future, separation-of-function 9-1-1 mouse

mutants could be used to clarify precisely how the 9-1-1 complexes mediate meiotic processes such as homolog synapsis, cohesion, and silencing. Moreover, continued genetic and biochemical analysis of the paralogs RAD9B and HUS1B holds promise for resolving the differential and overlapping roles of the canonical and alternative 9-1-1 complexes in spermatogenesis.

# Materials and methods

**Key resources table**

| Reagent type (species) or resource | Designation | Source or reference | Identifiers | Additional information |
|---|---|---|---|---|
| | | | | IF (1:100) |
| Antibody | Anti-RAD1; HM454 (rabbit polyclonal) | *Lyndaker et al., 2013a* | | WB (1:1000) |
| Antibody | Anti-RAD9A; HM456 (rabbit polyclonal) | *Lyndaker et al., 2013b* | | IF (1:100) |
| Antibody | Anti-RAD9B (rabbit polyclonal) | *Pérez-Castro and Freire, 2012* | | IF (1:100) |
| Antibody | Anti-phospho-histone H2A.X (Ser139) antibody, clone JBW301 (mouse monoclonal) | Millipore | Cat# 05-636; RRID:AB_309864 | IF (1:1000) |
| Antibody | SCP3 antibody [Cor 10G11/7] (mouse polyclonal) | Abcam | Cat# ab97672; RRID:AB_10678841 | IF (1:1000) |
| Antibody | Anti-SYCP3 (rabbit polyclonal) | *Lenzi et al., 2005* | | IF (1:1000) |
| Antibody | Rabbit anti-SCP1 polyclonal antibody, unconjugated | Abcam | Cat# ab15090; RRID:AB_301636 | IF (1:1000) |
| Antibody | Anti-Rad51 (Ab-1) rabbit pAb antibody (rabbit polyclonal) | Millipore | Cat# PC130; RRID:AB_2238184 | IF (1:1000) |
| Antibody | Anti-RPA2; UP2436 (rabbit polyclonal) | *Shi et al., 2019* | | IF (1:500) |
| Antibody | Anti-MEIOB; UP2327 (rabbit polyclonal) | *Luo et al., 2013* | | IF (1:500) |
| Antibody | Anti-replication protein A, clone RPA34-20 (mouse monoclonal) | Millipore | Cat# MABE285; RRID:AB_11205561 | IF (1:100) |
| Antibody | ATR antibody (rabbit polyclonal) | Cell Signaling | Cat# 2790; RRID:AB_2227860 | IF (1:100) |
| Antibody | Anti-TOBP1 (rabbit polyclonal) | *Rendtlew Danielsen et al., 2009* | | IF (1:500) |
| Antibody | Anti-phospho-Chk1 (ser317) (D12H3) XP (rabbit monoclonal) | Cell Signaling | Cat# 12302; RRID:AB_2783865 | IF (1:100) |
| Antibody | Anti-MLH1 (mouse monoclonal) | BD Biosciences | Cat# 550838; RRID:AB_2297859 | IF (1:1000) |
| Antibody | Anti-H1T (guinea pig polyclonal) | *Inselman et al., 2003* | | IF (1:500) |
| Antibody | Anti-HORMAD2; AB324 (rabbit polyclonal) | *Wojtasz et al., 2009* | | IF (1:500) |
| Antibody | Anti-HORMAD1; AB211 (rabbit polyclonal) | *Wojtasz et al., 2009* | | IF (1:500) |
| | | | | IF (1:100) |
| Antibody | Rabbit anti-SMC3 antibody, affinity purified (rabbit polyclonal) | Bethyl | Cat# A300-060A; RRID:AB_67579 | WB (1:1000) |

*Continued on next page*

Continued

| Reagent type (species) or resource | Designation | Source or reference | Identifiers | Additional information |
|---|---|---|---|---|
| Antibody | Rabbit anti-phospho SMC3 (S1083) IHC antibody (rabbit polyclonal) | Bethyl | Cat# IHC-00070; RRID:AB_2255076 | IF (1:100) <br><br> WB (1:1000) |
| Antibody | Anti-mouse TRA98 monoclonal antibody, unconjugated (mouse monoclonal) | BioAcademia | Cat# 73-003; RRID:AB_1056334 | IF (1:100) |
| Antibody | Rabbit anti-Lin28 polyclonal antibody, unconjugated (rabbit polyclonal) | Abcam | Cat# ab63740; RRID:AB_1310410 | IF (1:100) |
| Antibody | GAPDH monoclonal antibody (6C5) (mouse monoclonal) | Thermo Fisher Scientific | Cat# AM4300; RRID:AB_2536381 | WB (1:5000) |
| Antibody | β-Actin antibody (rabbit polyclonal) | Cell Signaling | Cat# 4967; RRID:AB_330288 | WB (1:5000) |
| Antibody | Goat anti-rabbit IgG (H + L) highly cross-adsorbed secondary antibody, Alexa Fluor 488 (rabbit polyclonal) | Thermo Fisher Scientific | Cat# A-11034; RRID:AB_2576217 | IF (1:1000) |
| Antibody | Goat anti-mouse IgG (H + L) antibody, Alexa Fluor 488 conjugated (mouse polyclonal) | Thermo Fisher Scientific | Cat# A-11017; RRID:AB_143160 | IF (1:1000) |
| Antibody | Goat anti-rabbit IgG (H + L) antibody, Alexa Fluor 594 conjugated (rabbit polyclonal) | Thermo Fisher Scientific | Cat# A-11012; RRID:AB_141359 | IF (1:1000) |
| Antibody | Goat anti-mouse IgG (H + L) highly cross-adsorbed secondary antibody, Alexa Fluor Plus 594 (mouse polyclonal) | Thermo Fisher Scientific | Cat# A32742; RRID:AB_2762825 | IF (1:1000) |
| Antibody | Goat anti-guinea pig IgG (H + L) highly cross-adsorbed secondary antibody, Alexa Fluor 647 (guinea pig polyclonal) | Thermo Fisher Scientific | Cat# A-21450; RRID:AB_141882 | IF (1:1000) |
| Sequence-based reagent | Cre ic318R | Lyndaker et al., 2013a | PCR primers | AGGGACACA GCATTGGAGTC |
| Sequence-based reagent | Cre ic202F | Lyndaker et al., 2013b | PCR primers | GTGCAAGCT GAACAACAGGA |
| Sequence-based reagent | Rad1 G1F | Wit et al., 2011 | PCR primers | AGGTACGTC AGTGCGATTACCCT |
| Sequence-based reagent | Rad1 G3R | Wit et al., 2011 | PCR primers | CCCTCAAGAT GTAACCTC ATCTAC |
| Sequence-based reagent | Hus1 3.107 | Lyndaker et al., 2013a | PCR primers | GGGCTGATGC GGAGGGTG CAGGTT |
| Sequence-based reagent | Hus1 Neo1 | Lyndaker et al., 2013b | PCR primers | GCTCTTTACT GAAGGCTCTTTAC |
| Sequence-based reagent | Hus1 5-OSMCS2 | Lyndaker et al., 2013a | PCR primers | GCGAAGACGG AATTGATCA GGCCACG |
| Sequence-based reagent | Hus1 5.-20 | Lyndaker et al., 2013b | PCR primers | CCGTCGGCCT GGTATCC GCCATGA |

Continued

| Reagent type (species) or resource | Designation | Source or reference | Identifiers | Additional information |
|---|---|---|---|---|
| Sequence-based reagent | Hus1 3.159 | *Lyndaker et al., 2013b* | PCR primers | CTCACAACTGCT ACAAGGTTAGGC |
| Commercial assay or kit | ApopTag Plus Peroxidase In Situ Apoptosis Kit | Millipore | Sigma-Aldrich: S7101 | |
| Chemical compound, drug | AZ20, ATR inhibitor | Selleckchem | Selleckchem: S7050 | |
| Software, algorithm | GraphPad Prism 9 | GraphPad | RRID:SCR_002798 | |

## Mice and genotyping

*Rad1* CKO and control mice in the 129Sv/Ev background were generated by crossing *Rad1^flox/flox^* mice with *Rad1^+/+^*, *Stra8-Cre^+^* mice to generate *Rad1^+/fl^*, *Stra8-Cre^+^* (*Rad1^+/-^*, *Stra8-Cre^+^*) mice. *Stra8-Cre* mice containing one null *Rad1* allele (*Rad1^+/-^*, *Stra8-Cre^+^*) were crossed with *Rad1^flox/flox^* mice to generate experimental germ-cell specific *Rad1* CKO mice (*Rad1^-/fl^*, *Stra8-Cre^+^*) and control mice (*Rad1^+/fl^*, *Stra8-Cre^+^*; *Rad1^+/fl^*, *Stra8-Cre^-^*; *Rad1^-/fl^*, *Stra8-Cre^-^*). *Rad1 flox* mice feature a conditional *Rad1* allele containing a K185R mutation that does not affect RAD1 function (*Wit et al., 2011*). *Hus1* CKO mice were used as previously reported (*Lyndaker et al., 2013a*). All mice used for this study were handled following federal and institutional guidelines under a protocol approved by the Institutional Animal Care and Use Committee (IACUC) at Cornell University. The Key resources table lists the genotyping primers used for mice in this study.

## Fertility tests

For fertility testing, 8- to 12-week-old *Rad1^-/fl^*, *Stra8-Cre+* and control males were singly housed with wild-type FVB females, where copulatory plugs were monitored daily. Once a plugged female was detected, the female was removed to a separate cage and monitored for pregnancy. Viable pups were counted on the first day of life.

## Epididymal sperm counts

Both caudal epididymides from 12-week-old mice were minced with fine forceps in 37°C in a Petri dish containing 1× phosphate buffered saline (PBS) and fixed in 10% neutral-buffered formalin (1:25 dilution). Sperm were counted using a hemacytometer and analyzed statistically using a Student's *t*-test between control and *Rad1* CKO mice.

## Treatment of mice with ionizing radiation or ATR inhibitor

For irradiation, control and *Rad1* CKO mice were placed in a $^{137}$Cesium-sealed source irradiator (J.L. Shepherd and Associates) with a rotating turntable and irradiated with 5 Gy IR. Testes were harvested for meiotic spreads 1 hr post radiation. For in vivo ATR inhibition, wild-type B6 mice were treated via oral gavage with 50 mg/kg AZ20 (Selleck Chemicals, S7050) reconstituted in 10% DMSO (Sigma), 40% propylene glycol (Sigma), and 50% water, and collected 4 hr later.

## Immunoblotting

Whole testis lysates from *Rad1* CKO and control mice were prepared in RIPA buffer (10 mM Tris-HCl, pH 8.0, 1 mM EDTA, 0.5 mM EGTA, 1% Triton X-100, 0.1% sodium deoxycholate, 0.1% SDS, 140 mM NaCl) supplemented with aprotinin, leupeptin, sodium orthovanadate, and phenyl-methylsulfonyl fluoride. Cell lysates were resolved by SDS-PAGE and immunoblotted using standard procedures. Bands were visualized on a VersaDoc MP 5000 Model (Bio-Rad) using a 1:1 ratio of WesternBright ECL Luminol/enhancer solution to WesternBright Peroxide Chemiluminescent peroxide solution (Advansta). Antibody information is provided in the Key resources table.

## Histology and immunohistochemistry

Testes were harvested from mice aged to 8 dpp, 4 weeks or 12 weeks of age. Testes were then fixed overnight in either Bouin's (RICCS Chemical) for H&E staining or 10% neutral-buffered formalin

(Fisher) for LIN28 (RRID:AB_1310410), TRA98 (RRID:AB_1056334), and TUNEL staining. Fixed testes were embedded in paraffin wax and sectioned at 5 μm. Immunofluorescence staining was used to detect LIN28 using rabbit polyclonal anti-LIN28 antibody (RRID:AB_1310410). Immunohistochemistry staining was used to detect TRA98 using rat monoclonal anti-TRA98 antibody (RRID:AB_1056334). TUNEL assay was performed using the Apoptag kit (EMD Millipore) as per the manufacturer's instructions. LIN28, TRA98, and TUNEL data were quantified in ImageJ by counting the number of positive cells per tubule for 50 tubules of each genotype for each age group. Differences between controls and *Rad1* CKOs were analyzed using Welch's unpaired *t*-test in GraphPad (RRID:SCR_002798). Staging of spermatocytes in stained sections was performed as described by others (*Ahmed and de Rooij, 2009*; *Meistrich and Hess, 2013*).

## Meiotic spreading and immunofluorescence staining

Meiotic spreads were prepared from 8- to 12-week-old mice as previously described (*Kolas et al., 2005*). Briefly, tubules from mice were incubated on ice in hypotonic extraction buffer for 1 hr. Tubules were then minced into single-cell suspension in 100 mM sucrose, and cells were spread on slides coated with 1% PFA with 0.15% TritionX-100 and incubated in a humidifying chamber for 4 hr or overnight. For immunostaining, slides were blocked using 10% goat serum and 3% BSA, followed by incubation overnight with primary antibody (listed in the Key resources table) at room temperature in a humidifying chamber. Secondary antibodies were incubated at 37°C for 2 hr in the dark, and slides were then coverslipped using anti-fade mounting medium (2.3% DABCO, 20 mM Tris pH 8.0, 8 μg DAPI in 90% glycerol). Meiotic chromosomal spreads were imaged with an AxioCam MRM using a Zeiss Imager Z1 microscope (Carl Zeiss, Inc) and processed with ZEN software (version 2.0.0.0; Carl Zeiss, Inc). Quantification of meiotic spreads was performed using Fiji for ImageJ. Statistical analysis was performed using Welch's unpaired *t*-test using GraphPad Prism9 (RRID:SCR_002798).

## RNA fluorescence in situ hybridization (RNA-FISH) and immunofluorescence staining

RNA-FISH was carried out with digoxigenin-labeled probe using BAC DNA, *Scml2*: RP24-204O18 (CHORI), and immunofluorescence using rabbit HORMAD2 antibody (gift from A. Toth) as previously described (*Mahadevaiah et al., 2009*). Images of RNA-FISH with immunofluorescence were captured using a Deltavision Microscopy System with a ×100/1.35 NA Olympus UPlanApo oil immersion objective.

## Orthology analysis

Human 9-1-1 subunit sequences were used to obtain their respective orthologs from Ensemble 101 (2020) and/or NCBI Gene from 33 representative mammalian species. Orthologs found in Ensemble having a ≥ 50% of both target and query sequence identity and a pairwise whole-genome alignment score of ≥50 were considered to have high confidence. Orthologs that did not meet those criteria were considered to have low confidence. Sequences only found in the NCBI Gene database were considered as high confidence if they were found to be syntenic. Synteny was determined based on whether the gene had at least one shared neighbor gene upstream or downstream that also was conserved. Species divergence across time was obtained from TimeTree website (http://timetree.org/).

## Phylogenetic analysis

Protein sequences of 9-1-1 orthologs were obtained using NCBI HomoloGene. Multiple alignment of protein sequences was done using Clustal Omega (1.2.2) implemented in Geneious Prime (2020.0.5). A substitution model was tested using ProtTest (v. 3.4.2). The selected substitution model with specific improvements was JTT + I + G + F (Jones–Taylor–Thornton;+ I: invariable sites; + G: rate heterogeneity among sites; + F: observed amino acid frequencies). Improvements were included to take account of any evolutionary limitations due to conservation of protein structure and function. A nonrooted phylogenetic tree was made using Maximum Likelihood interference (four gamma distributed rate) (*Nguyen et al., 2015*) and implemented with iTOL (itol.embl.de) (*Letunic and Bork, 2019*). Branch distance represents substitution rate, and branch support was performed with 1000 ultrafast bootstrap replicates. Nodes below 70% branch support were collapsed.

## ERC analysis

ERC calculations were completed using the ERC web tool at https://csb.pitt.edu/erc_analysis/ (*Wolfe and Clark, 2015*). Group analysis was performed to examine ERC values between all gene pairs indicated in *Figure 2A* and *Figure 5—figure supplement 1B and C* using UCSC gene sequences from 33 mammalian species as described in *Priedigkeit et al., 2015*. For *Figure 5—figure supplement 2A and B*, the protein set list for Gene Ontology subontology Meiosis I (GO:0007127) was obtained from AmiGO 2 (v2.5.13). ERC values were calculated against each of the 9-1-1 subunits using the ERC analysis website. Using R (v4.0.3), ERC values were depicted as a heatmap and a network plotted using the packages pheatmap (v1.0.12) and qgraph (v1.6.5), respectively. A cutoff of ERC value of 0.4 was used to determine significant comparisons. The Fruchterman and Reingold algorithm was used to generate a forced-directed layout to help determine clusters of highly connected nodes, and after 500 iterations the distance between nodes shows absolute edge weight (ERC values) between nodes.

## Acknowledgements

We are thankful to Dan Barbash and Eric Alani for helpful discussions and for providing critical feedback on the manuscript, to Mary Ann Handel and Attila Toth for providing reagents used in this study, and to Christina Jeon for early-stage contributions to the analysis of 9-1-1 subunit evolution. This work was supported in part by NIH grants R03 HD083621 (to RSW), R01 HD095296 (to MBS and RSW), R01 HD097987 (to PEC), NSF predoctoral fellowship DGE-1144153 (to CP), NIGMS 2R25GM096955 (to GAAM), and a National Center for Research Resources instrumentation grant (S10 RR023781). This work additionally was supported by the European Research Council (CoG 647971) and the Francis Crick Institute, which receives its core funding from Cancer Research UK (FC001193), UK Medical Research Council (FC001193), and Wellcome Trust (FC001193).

## Additional information

### Funding

| Funder | Grant reference number | Author |
| --- | --- | --- |
| National Institutes of Health | R03 HD083621 | Robert S Weiss |
| National Institutes of Health | R01 HD095296 | Robert S Weiss Marcus B Smolka |
| National Institutes of Health | R01 HD097987 | Paula Elaine Cohen |
| National Science Foundation | DGE-1144153 | Catalina Pereira |
| National Center for Research Resources | S10 RR023781 | Robert S Weiss |
| European Research Council | CoG 647971 | James Turner |
| Cancer Research UK | FC001193 | James Turner |
| Medical Research Council | FC001193 | James Turner |
| Wellcome Trust | FC001193 | James Turner |

The funders had no role in study design, data collection and interpretation, or the decision to submit the work for publication.

### Author contributions

Catalina Pereira, Conceptualization, Data curation, Formal analysis, Investigation, Methodology, Writing - original draft, Writing – review and editing; Gerardo A Arroyo-Martinez, Data curation, Formal analysis, Investigation, Methodology, Writing - original draft, Writing – review and editing; Matthew Z Guo, Michael S Downey, Shantha K Mahadevaiah, Charlton Tsai, Carl J Schiltz, Data

curation, Formal analysis, Investigation, Methodology; Emma R Kelly, Kathryn J Grive, Investigation, Methodology; Jennie R Sims, Data curation, Formal analysis, Investigation, Methodology, Writing – review and editing; Vitor M Faca, Data curation, Formal analysis; Niek Wit, Heinz Jacobs, Nathan L Clark, Raimundo Freire, Resources; James Turner, Funding acquisition, Project administration, Supervision; Amy M Lyndaker, Conceptualization, Writing – review and editing; Miguel A Brieno-Enriquez, Data curation, Formal analysis, Methodology, Writing – review and editing; Paula E Cohen, Conceptualization, Funding acquisition, Project administration, Resources, Supervision, Writing – review and editing; Marcus B Smolka, Conceptualization, Funding acquisition, Project administration, Supervision, Writing – review and editing; Robert S Weiss, Conceptualization, Funding acquisition, Project administration, Supervision, Writing - original draft, Writing – review and editing

### Author ORCIDs
Catalina Pereira http://orcid.org/0000-0003-3144-0909
Gerardo A Arroyo-Martinez http://orcid.org/0000-0003-2308-3286
Matthew Z Guo http://orcid.org/0000-0003-4741-7463
Michael S Downey http://orcid.org/0000-0001-9818-5274
Nathan L Clark http://orcid.org/0000-0003-0006-8374
Raimundo Freire http://orcid.org/0000-0003-4473-8894
James Turner http://orcid.org/0000-0003-1722-7677
Paula E Cohen http://orcid.org/0000-0002-2050-6979
Marcus B Smolka http://orcid.org/0000-0001-9952-2885
Robert S Weiss http://orcid.org/0000-0003-3327-1379

### Ethics
All mice used for this study were handled following federal and institutional guidelines under protocols approved by the Institutional Animal Care and Use Committee (IACUC) at Cornell University (protocol numbers 2011-0098 and 2004-0034).

### Decision letter and Author response
Decision letter https://doi.org/10.7554/eLife.68677.sa1
Author response https://doi.org/10.7554/eLife.68677.sa2

## Additional files

### Supplementary files
• Transparent reporting form

### Data availability
All data generated or analysed during this study are included in the manuscript and supporting file. Source Data files have been provided.

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
