## [Editor Report]

This paper nicely provides the roles of 9-1-1 checkpoint clamps (three DNA damage response clamps; Rad9A-Rad1-Hus1, Rad9B-Rad1-Hus1, Rad9B-Rad1-Hus1B) in mouse male meiosis, particularly DSB repair, chromosome synapsis, checkpoint signaling, and meiotic sex chromosome silencing.

---

## [Decision Letter]

**Decision letter after peer review:**

Thank you for submitting your article “Multiple 9-1-1 complexes promote homolog synapsis, DSB repair, and ATR signaling during mammalian meiosis” for consideration by *eLife*. Your article has been reviewed by 3 peer reviewers, one of whom is a member of our Board of Reviewing Editors, and the evaluation has been overseen by a Senior Editor. All acknowledge the potential of this paper. However, we agree that you need a major revision of your paper, particularly in respect of the co-submitted paper with Dr. Smolka.

Since using the same data set in co-submitted papers is unusual, we strongly ask you to move most of the ATR-dependent phosphor-proteomic data described in Figure 6 (and possibly Figure 7-since there is little insight on the role of ATR-dependent phosphorylation of Smc3) to the accompanying paper by Smolka. Importantly, to strengthen your conclusion in the paper and to differentiate from your original papers on Hus1 and Rad9a conditional knockout (cKO). We recommend you work on a more detailed characterization of Rad1 cKO. Below is the list of experiments and additional analysis for the revised version.

Essential revisions (experiments and analyses):

1. Since the role of Rad1, thus 9-1-1, in ATR-dependent meiotic sex chromosome inactivation (MSCI) is a very new observation, you need a more detailed description of defective MSCI in Rad1 cKO by staining with BRCA1. Moreover, Scml2 RNA FISH data are not sufficient to conclude MSCI defects. You should check the localization of RNA polymerase II to see if MSCI is disrupted.

2. Western blots in Figure 2C and Figure 7E are not appropriate because 12-week testes were used; these testes have a totally different cellular composition between controls and mutants. You should use juvenile testes, which have similar cellular composition between controls and mutants.

3. Analysis of the colocalization of RAD1 with RAD51, DMC1, RPA, MEIOB and/or CtIP. All combinations would be great. However, you can check some pairs of colocalization such as RAD51-RAD1, RPA2-RAD1, and CtIP-RAD1.

4. Quantification of the efficiency in Cre-mediated recombination by analyzing the frequency of round spermatids. In the text (line 242), 43% of spermatocytes are Rad1-negative. Does this mean that 57% of the cells are normal Rad1 localization? What percent of tubules have round spermatids?

5. In the same line, ~40% of spermatocytes show complete chromosome synapsis-these cells are positive for gH2AX and ATR staining (Figure 3D showed incomplete gH2AX domains, but Figure 5G showed an apparently normal gH2AX domain). The frequency of the phenotype needs to be scored. Given that ~ 40% of mutant cells completed chromosome synapsis, what is the ATR-activation phenotype in these ~ 40% of mutant cells that completed chromosome synapsis? Figure 3D showed incomplete gH2AX domains, but Figure 5G showed an apparently normal gH2AX domain. Please explain this discrepancy

6. Quantification of RAD1, RAD9A, and RA9B foci in different meiotic stages.

7. Provide a more accurate description of the location of 9-1-1 complexes as synapsis progresses. Are foci present in synapsed axis? (from the images provided in Figure 3A it seems so, but I don't think it's been described in the text)

8. No CHK1 phosphorylation sites were detected in supplemental table 1 of the phosphoproteomics data. However, the authors study pCHK1 in Figure 5F. Since CHK1 is not required for MSCI, and non-phosphor CHK1 has never been detected on the XY body, the validity of CHK1 data in Figure 5F is questionable. Western blots should be performed, at least in controls, to confirm the presence of pCHK1 (S317) in normal meiosis.

9. pSMC3 (1083) was not detected in supplemental table 1 of the phosphoproteomics data. Figure 7 does not add much information to the main story. This section needs additional clarification.

If possible:

1. Localization of Rad1 on chromosomes in Spo11 KO testis.

2. Defects in female meiosis of Rad1cKO. Alternatively, the localization of 9-1-1 on chromosomes in female meiosis.

Reanalysis of the results:

1. Quantification of the efficiency in Cre-mediated recombination by analyzing a frequency of round spermatids. In the text (line 242), 43% of spermatocytes are Rad1-negative. Does this mean that 57% of the cells are normal Rad1 localization? In the same line, ~40% of spermatocytes show complete chromosome synapsis-these cells are positive for gH2AX and ATR staining (Figure 3D showed incomplete gH2AX domains, but Figure 5G showed an apparently normal gH2AX domain).

2. Quantification of RAD1, RAD9A, and RA9B foci in different meiotic stages.

3. Provide a more accurate description of the location of 9-1-1 complexes as synapsis progresses. Are foci present in synapsed axis? (from the images provided in Figure 3A it seems so, but I don't think it's been described in the text)

*Reviewer #1:*

The paper by Pereira et al. describes the characterization of a testis-specific conditional knockout (CKO) of the Rad1 gene, which encodes a component of DNA damage response (DDR) sensor, 911 clamp (Rad9-Rad1-Hus1). Mammals have three distinct DDR clamps, Rad9A-Rad1-Hus1, Rad9B-Rad1-Hus1, and Rad9B-Rad1-Hus1B, in which Rad1 is a share component. The authors showed Rad1 CKO is defective in the repair of meiotic DNA double-strand breaks (DSBs), chromosome synapsis and meiotic sex chromosome inactivation (MSCI) mediated by ATR kinase. This is an extension of previous works by the authors on Hus1 and Rad9A CKO mice. Moreover, with phosphor-proteomic analysis of proteins in testis of the Rad1 CKO and ATR inhibitor-treated testis, the authors showed the link of Rad1- and ATR-dependent phosphorylation to various chromosomal events such as cohesion.

*Reviewer #2:*

This paper examined the function of RAD1, a common subunit of mammalian 9-1-1 complex in male meiosis. Importantly, the phenotype of Rad1 CKO appeared to be more severe than that of other redundant subunits of the 9-1-1 complex (Hus1 CKO and Rad9a CKO), and the authors presented promising pieces of data that show that ATR signaling was largely impaired. Pictures are high quality in general. However, a subset of mutant germ cells apparently complete meiotic prophase and reached round spermatids, and it appeared that ~ 40% of mutant cells completed chromosome synapsis (Figure 3- supplement 1D). Therefore, after reading this manuscript, it is not clear how many mutant cells showed such severe prophase defects (as demonstrated in some main figures); nor is the efficiency of Cre deletion confirmed. Although the study has potential, it is somewhat preliminary to conclude the mechanism by which the 9-1-1 complex regulates meiosis.

*Reviewer #3:*

In this study, the Weiss lab reveals the function of RAD1 during the mouse meiotic prophase. RAD1 is a component of the 9-1-1 complexes, which are responsible to activate the DNA damage response kinase ATR in somatic cells. Canonical (9A-1-1) and alternative (9B-1-1 and 9B-1-1B) 9-1-1 complexes have been described in mice. Importantly, all 9-1-1 complexes contain RAD1. Previous work from the group studying RAD9A suggested that the canonical complex had no major role in meiosis. Nonetheless, nothing was known about the function of the rest of the 9-1-1 complexes in meiosis. The presented data clearly shows that RAD1 is required to complete meiotic recombination, homologous synapsis, meiotic sex chromosome inactivation, and to activate ATR in spermatocytes. In general, the study is well designed and executed, the manuscript is well organized and reads easily, and the conclusions are supported by the data.

This study highlights the importance of the 9-1-1 complexes and the ATR signaling pathway in spermatocytes, something that was not known before. The data presented here will be very useful for the DNA repair, meiosis, and reproductive biology communities studying the roles of the ATR signaling pathway. However, I think the study could benefit from a more thorough analysis of the presence of the different 9-1-1 complexes in mouse testis as well as the relation of the 9-1-1 complex with early recombination markers. Also, I missed the description of the role of RAD1 in female mice fertility, something that would clearly expand the relevance of the findings described in the paper.

---

## [Author Response]

Since using the same data set in co-submitted papers is unusual, we strongly ask you to move most of the ATR-dependent phosphor-proteomic data described in Figure 6 (and possibly Figure 7-since there is little insight on the role of ATR-dependent phosphorylation of Smc3) to the accompanying paper by Smolka. Importantly, to strengthen your conclusion in the paper and to differentiate from your original papers on Hus1 and Rad9a conditional knockout (cKO). We recommend you work on a more detailed characterization of Rad1 cKO. Below is the list of experiments and additional analysis for the revised version.

We appreciate these recommendations on how best to improve our manuscript and more effectively align it with the co-submitted resource paper. As suggested, we have removed the phosphoproteomic data (original Figure 6) from the revised manuscript. We retained the analysis of SMC3 phosphorylation as an example of a RAD1- and ATR-dependent phosphorylation event that is consistent with the *Rad1* CKO phenotypes we report, but we have de-emphasized this result by moving several of the related panels to a supplemental figure (Figure 5—figure supplement 1) and by integrating the results into a section that describes several different ATR signaling defects (revised Figure 5) rather than giving it a separate section. The Results section now ends with a figure devoted to the important new findings of meiotic silencing defects in *Rad1* CKO mice (revised Figure 6).

Essential revisions (experiments and analyses):1. Since the role of Rad1, thus 9-1-1, in ATR-dependent meiotic sex chromosome inactivation (MSCI) is a very new observation, you need a more detailed description of defective MSCI in Rad1 cKO by staining with BRCA1. Moreover, Scml2 RNA FISH data are not sufficient to conclude MSCI defects. You should check the localization of RNA polymerase II to see if MSCI is disrupted.

To further characterize the MSCI defect, we performed additional staining for BRCA1 and RNA Pol II in control and *Rad1* CKO spermatocytes (Figure 6). In pachytene-like cells lacking RAD1 expression, BRCA1 showed a defective localization pattern similar to what we observed for TOPBP1 and ATR, with BRCA1 failing to localize to all unsynapsed regions in *Rad1* CKO spermatocytes (Figure 6D).

Whereas RNA Pol II showed the expected exclusion from the sex body in pachytene spermatocytes from control males, RNA Pol II was diffusely distributed in RAD1-deficient pachytene-like spermatocytes with extensive asynapsis. However, the only way to definitively establish whether a protein has a primary role at the XY pair to initiate MSCI is to specifically assay cells in which autosomal synapsis is unaffected, because asynapsis of the autosomes antagonizes MSCI by sequestering the proteins that are required for silencing away from the sex chromosomes (DOI: 10.1083/jcb.200710195). We therefore further examined apparently normal, fully synapsed cells from *Rad1* CKO testes and observed a higher frequency of abnormal localization of RNA Pol II in the sex body (27.8 ± 19.1%) as compared to control cells (8.8 ± 2.4% of cells; Figure 6E). Together with the defective localization of gH2AX and the RNA FISH data showing aberrant X-linked gene expression in cells from *Rad1* CKO testes with apparently normal synapsis, these new results provide further evidence that 9-1-1 complexes are important for MSCI.

2. Western blots in Figure 2C and Figure 7E are not appropriate because 12-week testes were used; these testes have a totally different cellular composition between controls and mutants. You should use juvenile testes, which have similar cellular composition between controls and mutants.

We agree that juvenile testes provide a more similar cellular composition for comparing control and mutant mice in total testis lysates. In order to address this, we performed immunoblotting on testes from 14 day old mice and observed decreased RAD1 levels in *Rad1* CKO samples (Figure 2—figure supplement 1), similar to the reduction detected in samples from adult *Rad1* CKO testes (Figure 2C).

Immunoblotting for SMC3 and pSMC3 (1083) in P14 testis lysates was also performed. Consistent with the original immunoblot of whole testes from 12 week-old mice (Figure 5—figure supplement 1D), we observed reduced levels of pSMC3 (1083) in *Rad1* CKO testes from 14 day old mice (Figure 5—figure supplement 1E).

3. Analysis of the colocalization of RAD1 with RAD51, DMC1, RPA, MEIOB and/or CtIP. All combinations would be great. However, you can check some pairs of colocalization such as RAD51-RAD1, RPA2-RAD1, and CtIP-RAD1.

We previously reported that RAD51 and RAD9A co-localize on meiotic chromosome cores (DOI: 10.1371/journal.pgen.1003320). In addition, Freire et al. previously reported that RAD1 and DMC1 co-localize in immunofluorescence assays and thus are in the same general vicinity but actually have distinct localizations by immunoelectron microscopy (DOI: 10.1101/gad.12.16.2560). We have fortified our descriptions of these prior results and focused new experimentation on co-staining of RPA and RAD1. We report that relatively limited co-localization in early Prophase I (Figure 4 Supplement 1A). As RPA and RAD1 foci become apparent along chromosome cores, additional overlap in signal is evident, although much of the signal is non-overlapping which is not surprising given that RPA coats single-stranded DNA whereas 9-1-1 likely is present on double-stranded DNA. Co-localization of RAD1 and RPA on autosomes in pachytene-stage cells may in part reflect recombination intermediates during the final stages of DSB repair (DOI:10.1016/j.molcel.2020.06.015). In late pachytene, RAD1 is known to coat the unsynapsed regions of the X and Y, and is present there even when there are no apparent RPA foci, which likely reflects the role of RAD1 in MSCI rather than DSB repair. Additional data from our studies and others hint at 9-1-1 functions in synapsis and cohesion, and these roles also may position RAD1 independently of RPA throughout Prophase I.

4. Quantification of the efficiency in Cre-mediated recombination by analyzing the frequency of round spermatids. In the text (line 242), 43% of spermatocytes are Rad1-negative. Does this mean that 57% of the cells are normal Rad1 localization? What percent of tubules have round spermatids?

We appreciate this comment from the reviewer and agree that a clearer description of the extent of RAD1 loss in *Rad1* CKO mice was needed. As noted by the reviewer, RAD1 expression was undetectable in 43% of spermatocytes. Additional data further suggest that among the remaining 57% of spermatocytes with detectable RAD1 expression, a subset of the cells had functional defects reflecting significant but partial loss of RAD1 expression. First, approximately 60% of *Rad1* CKO meiocytes had synapsis defects. Additionally, some of the cells with apparently normal synapsis had other functional defects. Namely, 15% of *Rad1* CKO spermatocytes with apparently complete synapsis exhibited defects in γH2AX staining on the XY body, 28% showed aberrant RNA Pol II localization, and 29% had defects in silencing of the X-linked *Scml2* gene. We have added a new figure (Figure 3—figure supplement 1B) that summarizes the three categories of spermatocytes observed in *Rad1* CKO mice: (1) those with no detectable RAD1 foci and severe defects; (2) those with some detectable RAD1 foci but clear functional defects; and (3) those with detectable RAD1 foci and no apparent functional defects. In total, approximately 72% of *Rad1* CKO cells had functional defects.

Our new quantification of round spermatids in testis sections from 12-week-old mice further supports the notion that the majority of *Rad1* CKO meiocytes had functional defects. Although the severe germ cell loss in *Rad1* CKO mice prevented staging of tubules, we found that 65% of tubules in *Rad1* CKO testes had fewer than 10 round spermatids while no tubules from control had fewer than 10 round spermatids. We have updated the text to include this important point.

5. In the same line, ~40% of spermatocytes show complete chromosome synapsis-these cells are positive for gH2AX and ATR staining (Figure 3D showed incomplete gH2AX domains, but Figure 5G showed an apparently normal gH2AX domain). The frequency of the phenotype needs to be scored. Given that ~ 40% of mutant cells completed chromosome synapsis, what is the ATR-activation phenotype in these ~ 40% of mutant cells that completed chromosome synapsis? Figure 3D showed incomplete gH2AX domains, but Figure 5G showed an apparently normal gH2AX domain. Please explain this discrepancy

In the original manuscript, Figure 3D showed asynapsed autosomes with incomplete gH2AX localization. Original Figure 5G, from an RNA FISH experiment, showed sex chromosomes with gH2AX localizing only to a portion of the X and absent from the Y, an abnormal pattern that correlated with the lack of silencing of the X-linked gene *Smcl2*.

To quantify the frequency of aberrant gH2AX localization, we further analyzed gH2AX staining at the sex body in both control and *Rad1* CKO cells with apparently normally synapsis. Abnormal staining was defined as either lack of gH2AX on the XY or extension of the gH2AX domain beyond the XY onto a nearby autosome. This quantification revealed abnormal gH2AX staining at the XY in 3.2 ± 1.5%, of pachytene-stage control cells. However, in *Rad1* CKO mice, 15.1 ± 11.5%, of cells with apparently normal synapsis had gH2AX defects at the XY (Figure 3E and Figure 3—figure supplement 2C). A fraction of *Rad1* CKO cells with apparently normal synapsis also showed defective RNA Pol II localization (27.8 ± 19.1%), and aberrant *Smcl2* expression was observed at a similar frequency (28.9 ± 3.2%). We conclude that among the 40% of cells in the *Rad1* CKO model with apparently complete synapsis, a subset of cells have a partial reduction of RAD1 that results in functional defects in ATR signaling and silencing. The new schematic shown in Figure 3—figure supplement 1B highlights this population of cells with RAD1 foci present but clear functional defects.

6. Quantification of RAD1, RAD9A, and RA9B foci in different meiotic stages.

We have quantified RAD1, RAD9A and RAD9B foci in control cells and added these numerical data to the paper to provide a clearer picture of how 9-1-1 subunits are distributed throughout meiotic prophase I. We found that RAD1 is present at high levels in early prophase (209.3 ± 23.9 RAD1 foci in leptonema and 208.2 ± 9.15 in zygonema) and reduced as the cells reached mid-pachynema (120.8 ± 27.3 RAD1 foci). Foci counts for RAD9A and RAD9B followed a similar trend as RAD1 in leptotene-stage cells (145.5 ± 20.8 RAD9A foci; 211.5 ± 50.8 RAD9B foci) and zygotene-stage cells (125.2 ± 25.6 RAD9A foci; 230.5 ± 40.5 RAD9B foci) and also dropped by mid-pachynema (107.3 ± 32.6 for RAD9A; 125.5 ± 19.2 for RAD9B). In pachytene-stage cells we observed RAD1 coating of the XY cores while RAD9A and RAD9B localized to the XY as discrete foci. Whether the differences in distribution of 9-1-1 subunits along the sex chromosomes reflect technical limitations of current immunoreagents or distinct roles for 9-1-1 subunits in DNA repair, MSCI or other processes is an open question for future investigation.

7. Provide a more accurate description of the location of 9-1-1 complexes as synapsis progresses. Are foci present in synapsed axis? (from the images provided in Figure 3A it seems so, but I don't think it's been described in the text)

We thank the reviewers for pointing out that our description of 9-1-1 localization relative to homolog synapsis was underdeveloped. As DSBs occur and repair begins in leptonema, RAD1 foci were present in high numbers. As synaptonemal complex formation progressed in zygonema, RAD1 colocalized with axial elements on the chromosome cores. The reviewer is correct that in mid-pachynema RAD1 remained present on fully synapsed regions of the autosomes. RAD1 signal was also highly concentrated on the XY cores. RAD1 continued to be observed on the XY cores in late-pachynema but was no longer present on autosomes. We have discussed these patterns more explicitly in the revised text and suggest that the pattern reflects several possible functional roles for 9-1-1 complexes in synapsis, cohesion, DSB repair, and MSCI as detailed in the manuscript.

8. No CHK1 phosphorylation sites were detected in supplemental table 1 of the phosphoproteomics data. However, the authors study pCHK1 in Figure 5F. Since CHK1 is not required for MSCI, and non-phosphor CHK1 has never been detected on the XY body, the validity of CHK1 data in Figure 5F is questionable. Western blots should be performed, at least in controls, to confirm the presence of pCHK1 (S317) in normal meiosis.

The reviewer is correct that CHK1 was not identified in the companion phosphoproteomic screen as a RAD1- and ATR-dependent target. However, in somatic cells the regulation of CHK1 phosphorylation via the 9A-1-1/TOPBP1/ATR axis is well established. Given our observations of defective ATR localization in *Rad1* CKO spermatocytes, we endeavored to examine CHK1 phosphorylation levels.

While CHK1 does not directly participate in MSCI, it has been implicated in meiotic DNA damage signaling and meiotic progression. In addition, Fedoriw et al. (DOI: 10.1242/dev.126078) previously reported that CHK1 and pCHK1 S317 localize to the XY core and XY chromatin loops. This is in line with what we observe in our control spermatocytes stained for pCHK1 S317 (Figure 5C-D). We did not intend to conclude that the CHK1 phosphorylation and MSCI defects we observed in RAD1-deficient cells were inter-connected, and therefore we have moved the CHK1 data to the figure describing characteristics of the TOPBP1/ATR signaling axis (Figure 5). At the suggestion of the reviewer, we also performed pCHK1 immunoblotting on whole testis lysates. Consistent with our other results, *Rad1* CKO testes showed a reduction in pCHK1 (S317) and pCHK1 (S345) relative to control testes (Figure 5—figure supplement 1A). Together, the immunostaining and immunoblotting data indicate that RAD1 is required for CHK1 S317 phosphorylation during meiosis.

9. pSMC3 (1083) was not detected in supplemental table 1 of the phosphoproteomics data. Figure 7 does not add much information to the main story. This section needs additional clarification.

Three 3 phospho-SQ sites on SMC3 (787, 1067, 1083) were detected in the phosphoproteomics data set, but only 787 was significantly reduced in ATRi and *Rad1* CKO samples. Nevertheless, given that phosphorylation of SMC3 (1083) has been shown previously to be linked to ATR signaling and there is a commercial antibody readily available to us, we decided to interrogate the site further. The lack of pSMC3 (1083) phosphorylation that we observed in *Rad1* CKO spermatocytes is intriguing since it hints at a role for the 9-1-1 complexes in cohesin regulation via ATR phosphorylation, with potential implications for the DSB repair and synapsis defects we observe. However, we agree that the role of 911-mediated SMC3 regulation remains to be determined, and therefore we have placed less emphasis on this finding in the revised manuscript and include it as one of several examples in revised Figure 5 of altered phosphorylation of ATR targets in *Rad1* CKO mice.

If possible:1. Localization of Rad1 on chromosomes in Spo11 KO testis.

We attempted this experiment and observed significantly reduced RAD1 loading in *Spo11* KO spermatocytes, but unfortunately we could not unambiguously determine whether RAD1 focus formation was completely disrupted by SPO11 loss or if some residual focus formation occurred. We therefore elected not to include these results. However, we have added a note on our previously reported finding that localization of the canonical RAD9A subunit on meiotic chromosomes relies on DSB formation by SPO11 (DOI: 10.1371/journal.pgen.1003320).

2. Defects in female meiosis of Rad1cKO. Alternatively, the localization of 9-1-1 on chromosomes in female meiosis.

We agree that understanding the roles of the 9-1-1 complexes in female meiosis is of great interest. Unfortunately, the *Stra8-Cre* transgene used to disrupt *Rad1* in our studies is not active in females. We are currently exploring other approaches to target the 9-1-1 complexes during female meiosis.